

# A study of long-range transported smoke aerosols in the Upper Troposphere/Lower Stratosphere

Qiaoyun Hu[1], Philippe Goloub[1], Igor Veselovskii[2], Juan-Antonio Bravo Aranda[3], Ioana Popovici[1,4],
Thierry Podvin[1], Martial Haeffelin[3], Anton Lopatin[5], Christophe Pietras[3], Xin Huang[5],
Benjamin Torres[1], and Cheng Chen[1]

[1]Univ. Lille, CNRS, UMR8518 – LOA – Laboratoire d'Optique Atmosphérique, 59000 Lille, France
[2]Physics Instrumentation Center of GPI, Troitsk, Moscow, 142190, Russia
[3]Institut Pierre Simon Laplace, École Polytechnique, CNRS, Université Paris-Saclay, 91128 Palaiseau, France
[4]CIMEL Electronique, 75011 Paris, France
[5]GRASP-SAS, Remote sensing developments, 59650 Villeneuve d'Ascq, France

**Correspondence:** Qiaoyun Hu (qiaoyun.hu@univ-lille.fr)

**Abstract.** Long-range transported smoke aerosols in the UTLS (Upper Troposphere/Lower Stratosphere) over Europe were detected in Summer 2017. The measurements of ground-based instruments and satellite sensors indicate that the UTLS aerosol layers were originated from Canadian wildfires and were transported to Europe by UTLS advection. In this study, the observations of two multi-wavelength Raman Lidar systems in northern France (Lille and Palaiseau) are used to derive aerosol

properties, such as optical depth of the UTLS layer, Lidar ratios at 355 and 532 nm and particle linear depolarization ratios at 355, 532 and 1064 nm. The optical depth of the UTLS layers at 532 nm varies from 0.05 to above 0.20, with very weak spectral dependence between 355 and 532 nm. Lidar ratios at 355 nm are in $31 \pm 15$ sr to $45 \pm 9$ sr range and at 532 nm, the Lidar ratios are in the range of $54 \pm 12$ sr to $58 \pm 9$ sr. Such spectral dependence of Lidar ratio is known to be a characteristic feature of aged smoke. The typical particle depolarization ratios in the UTLS smoke layer are $25 \pm 4\%$ at 355 nm, $19 \pm 3\%$ at

532 nm and $4.5 \pm 0.8\%$ at 1064 nm. The relatively high depolarization ratios and such spectral dependence are an indication of a complicated morphology of aged smoke particles. We found an increase of depolarization ratio versus transport time. The depolarization ratio at 532 nm increases from below $2 - 5\%$ for fresh smoke to over 20% for smoke aged more than 20 days. The $3\beta + 2\alpha$ observations of two cases at Palaiseau and Lille sites were inverted to the aerosol microphysical properties using *regularization* algorithm. The particles distribute in the 0.1–1.0 $\mu$m range with effective radius of $0.33 \pm 0.10$ $\mu$m for

both cases. The derived complex refractive indices are $1.52(\pm 0.05) + i0.021(\pm 0.010)$ and $1.55(\pm 0.05) + i0.028(\pm 0.014)$ for Palaiseau and Lille data. The retrieved aerosol properties were used to calculate the direct radiative forcing (DRF) effect specific to the UTLS aerosol layers. The simulations derive daily net DRF efficiency of -79.6 $\mathrm{Wm^{-2}\tau^{-1}}$ at the bottom of the atmosphere for Lille observations. At the top of the atmosphere, the net DRF efficiency is -7.9 $\mathrm{Wm^{-2}\tau^{-1}}$. The results indicate that the UTLS aerosols strongly reduce the radiation reaching the terrestrial surface by absorption. The heating rate of the

UTLS layers is estimated to be 3.7 $K\,day^{-1}$. The inversion of Palaiseau data leads to similar results. The heating rate predicts a temperature increase within the UTLS aerosol layer, which has been observed by the radiosonde temperature measurements.



# 1 Introduction

UTLS lies between the middle troposphere and the middle stratosphere. UTLS aerosols play an important role in the global radiative budget and chemistry-climate coupling (Deshler, 2008; Kremser et al., 2016; Shepherd, 2007). According to the in-situ measurements provided by PALMS (Particle Analysis by Laser Mass Spectrometry) instrument (Murphy et al., 2007, 2014),

UTLS aerosols are categorized into three branches: sulphuric acid with metals from ablation of meteoroids, nearly sulphuric acid with associated water and organic-sulphate particles, as well as a small fraction of dust, sea salt and other types originating from the troposphere. The volcanic eruption is considered as the most significant contribution of stratospheric aerosols because the explosive force could be sufficient enough to penetrate the tropopause, which is regarded as a barrier to the convection between the troposphere and stratosphere.

Lidar is an important tool in studying and profiling the UTLS aerosols. Hofmann et al. (2009) presented an increase of background stratospheric aerosols observed by Lidar system. The study showed the aerosol backscattering in the altitude range 20–30 km increased by 4–7% per year. Khaykin et al. (2017) studied the variability and evolution of the midlatitude stratospheric aerosols using 22 years of ground-based Lidar and satellite observations. This study provided an indication of a growth in the non-volcanic component of stratospheric aerosol over the last two decades. Zuev et al. (2017) presented 30-year Lidar

observations of the stratospheric aerosol layers coming from volcanic eruptions, polar stratospheric clouds and strong wildfire emissions over western Siberia.

In the summer of 2017, intense wildfires have spread in the west and north of Canada. By mid-August, the burnt area had grown to almost 9000 km$^2$ in British Columbia, which broke the record set in 1958 (see the link). The severe wildfires generated very strong pyro-cumulonimbus clouds (see the link) which were recorded by the satellite imaginary MODIS (Moderate Resolu-

tion Imaging Spectrometer). The pyro-cumulonimbus clouds are an extreme form of the pyro-cumulus clouds which have the potential to transport the fire emissions from the planetary boundary layer to the UTLS (Luderer et al., 2006; Trentmann et al., 2006), thus affecting the atmospheric chemistry of the stratosphere. Previous studies have (Fromm et al., 2000; Fromm and Servranckx, 2003) reported smoke aerosols lofted to the UTLS. The GOES-15 (Geostationary Operational Environmental Satellite) detected five pyro-cumulonimbus clouds over the British Columbia on 12 August 2017 (see the link).

Reoccurring aerosol layers in the UTLS were detected by the Lidar systems in northern France during 19 August and 12 September 2017. In this study, we present the Lidar observations from two French Lidar stations: Lille (50.612°N, 3.142°E, 60 m a.s.l) and Palaiseau (48.712°N, 2.215°E, 156 m a.s.l). Satellite measurements from multiple sensors, including UVAI (Ultraviolet aerosol index) from the OMPS NM (Ozone Mapping and Profiler Suite, Nadir Mapper), CO concentration from AIRS (Atmospheric Infrared Sounder), backscatter coefficient and depolarization profiles from CALIPSO (Cloud-Aerosol Lidar and

Infrared Pathfinder Satellite Observations) help identify the source and the transport pathway of the UTLS layers. We focus on the retrieval of the aerosol optical and microphysical properties from the Lidar measurements. Further, we study the radiative effect of the UTLS aerosol layer.



## 2 Ground-based and satellite observations

### 2.1 Simultaneous Lidar and sun photometer observations

LILAS is a multi-wavelength Raman Lidar (Bovchaliuk et al., 2016; Veselovskii et al., 2016) operated at LOA (Laboratoire d'Optique Atmosphérique, Lille, France). LILAS system is transportable and has three elastic channels (355, 532 and 1064

nm), with the capability of measuring the depolarization ratios at three wavelengths and three Raman channels: 387, 408 and 530 nm. IPRAL system (Bravo-Aranda et al., 2016; Haeffelin et al., 2005) is a multi-wavelength Raman Lidar operated at SIRTA (Site Instrumental de Recherche par Télédétéction Atmosphérique, Palaiseau, France). The distance between the two systems is around 300 km. Lidar IPRAL has the same elastic channels, but the three Raman channels are 387, 408 and 607 nm. In IPRAL system, the depolarization ratio is only measured at 355 nm. The two Lidar systems were operated independently

and both observed reoccurring smoke layers in the UTLS during the period of 19 August to 12 September 2017. In addition, sun photometer measurements are available at Lille and Palaiseau, which are both affiliated stations of AERONET (Aerosol Robotic Network). LILAS and IPRAL Lidar systems are affiliated to EARLINET (European Aerosol Research Lidar Network) (Bösenberg et al., 2003; Böckmann et al., 2004; Matthais et al., 2004; Papayannis et al., 2008). Both systems perform regular measurements and follow the standard EARLINET data quality check and calibration procedures.

On 29 August, three Lidar systems in northern France simultaneously observed an UTLS aerosol layer. The three Lidar systems are LILAS, IPRAL and a single wavelength (532 nm) CIMEL micro-pulse Lidar, which is set up in a light mobile system, MAMS (Mobile Aerosol Monitoring System, Popovici et al. (2018)) to explore aerosol spatial variability. MAMS was traveling between Palaiseau and Lille on 28 and 29 August. MAMS is equipped with a mobile sun photometer, PLASMA (Karol et al., 2013), capable to measure columnar aerosol optical depth (AOD) along the route.

Figure 1 shows the normalized Lidar range-corrected signals and columnar AOD at 532 nm derived from sun photometer measurements on 29 August 2017. The aerosol layers in the UTLS, stretching from 16 to 20 km, were detected by the three Lidars. The IPRAL Lidar system in Palaiseau detected the aerosol layer in the range of 16–20 km on 29 August. The columnar AOD showed no significant variations, staying between 0.30 and 0.40, from 1000 UTC to 1600 UTC and started decreasing from 1700 UTC. Along the route Palaiseau-Lille, MAMS Lidar observed a layer between 16 and 20 km consisting of two well-

separated layers. The columnar AOD was very stable, around 0.40, all along the route from Palaiseau to Lille. Lidar LILAS in Lille observed a shallow layer between 18–20 km at about 0800 UTC on 29 August. The thickness of the layer increased to 4 km until 1600 UTC. The columnar AOD increased from 0.20 to 0.40 from 0800 UTC to 1400 UTC. If we assume the contribution of tropospheric AOD does not change during 0800 UTC and 1400 UTC, the increased optical depth, 0.2, comes from the contribution of the UTLS aerosol layer.



## 2.2 Radiosonde measurements

Fromm et al. (2005, 2008) showed an increase of temperature in the stratospheric smoke layers. This "warming" effect has been observed in aged absorbing aerosols. We take the radiosonde measurements from two stations closest to the Lidar sites: Trappes (48.77°N, 1.99°E, France) and Beauvechain (50.78°N, 4.76°E, Belgium). Trappes is about 20 km to Palaiseau and Beauvechain is 120 km to Lille. Considering the large spatial distribution of the UTLS aerosols, it is likely that the radiosonde passed through this UTLS layer. Figure 2 shows the temperature at 0000, 1200 UTC, 29 August for Trappes and 2100 UTC, 29 August for Beauvechain. To compare, we plot the temperature profile of Trappes at 1200 UTC, 21 August, when no UTLS aerosol layer presented. The temperature profiles show clearly an enhancement between 16 and 20 km, consistent with the altitude of the UTLS aerosol layers. In spite of the spatial and temporal displacements of Lidar and radiosonde measurements, the coincidence of the temperature profiles between Trappes and Beauvechain suggests that the UTLS aerosol layer was widely distributed and the Lidar and radiosonde both observed the UTLS layer.

## 2.3 MODIS measurements

MODIS is a key instrument onboard the Terra (originally known as EOS AM-1) and Aqua (originally known as EOS PM-1) satellites. Terra MODIS and Aqua MODIS are viewing the entire Earth's surface every 1 to 2 days. Several episodes of Canadian wildfires have been observed since early July 2017. On 12 August, MODIS observed a thick, grey plume arising from the British Columbia in the west of Canada (not shown but avalaible on the webpage of WorldView). Figure 3 shows the corrected surface reflectance overlaid with the fires and thermal anomalies on 15 August 2017. The region marked with the green dashed line takes on a grey-yellow colour and in its southwest, MODIS detected a belt of fire spots. Additionally, during the week of 13–19 August, MODIS (see WorldView) observed a widespread cloud coverage over Canada and showed that clouds layers were overshadowed by the smoke plumes, meaning that the plumes were lofted above the cloud layers, as shown in Figure 3.

## 2.4 OMPS NM UVAI maps

UVAI is a widely used parameter in characterizing UV-absorbing aerosols, such as desert dust, carbonaceous aerosols coming from anthropogenic biomass burning, wildfires and volcanic ash. The UVAI is determined using the 340 and 380 nm wavelength channels and is defined as:

$$UVAI = -100 \times \left\{ \log_{10} \left[ \frac{I_{340}}{I_{380}} \right]_{meas} - \log_{10} \left[ \frac{I_{340}}{I_{380}} \right]_{calc} \right\} \tag{1}$$

where $I_{340}$ and $I_{380}$ are the backscattered radiance at 340 and 380 nm channel. The subscript $meas$ represents the measurements and the $calc$ represents the calculation using a radiative transfer model for pure Rayleigh atmosphere. The UVAI is defined so that positive values correspond to UV-absorbing aerosols and negative values correspond to non-absorbing aerosols (Hsu et al., 1999). The OMPS NM onboard the Suomi NPP (National Polar-orbiting Partnership) is designed to measure the total column ozone using backscattered UV radiation between 300–380 nm. A 110° field-of-view (FOV) telescope enables



full daily global coverage (McPeters et al., 2000; Seftor et al., 2014). Figure 4 shows the evolution of UVAI from OMPS NM (Jaross, 2017) every two days during 11 and 29 August 2017. A plume with relatively high UVAI first occurred over the British Columbia on 11 August, and the intensity of the plume was moderate. From 13 to 17 August, the plume spread in the northwest-southeast direction and the UVAI in the centre of the plume reached above 10. On 19 August, the plume centre

reached the Labrador Sea and the front of the plume reached Europe. From 21 to 29 August, the UVAI in the map is much lower than the previous week. During this period, we can still distinguish a plume propagating eastward from the Atlantic to Europe, with the UVAI damping during the transport. Figure 4(e)–(f) show that Europe was overshadowed by the high-UVAI plume during 19 and 29 August.

## 2.5   AIRS CO maps

AIRS is a continuously operating cross-track scanning sounder onboard NASA's Aqua satellite launched in May 2002. AIRS covers the 3.7 to 16 $\mu$m spectral range with 2378 channels and a 13.5 km nadir FOV (Susskind et al., 2014; Kahn et al., 2014). The daily coverage of AIRS is about 70% of the globe. AIRS is designed to measure the water vapour and temperature profiles. It includes the spectral features of the key carbon trace gases, $CO_2$, $CH_4$ and CO (Haskins and Kaplan, 1992). The CO, as a product of the burning process, can be taken as a tracer of biomass burning aerosols due to its relatively long lifetime of 1/2 to

3 months. The current CO product from AIRS is very mature because the spectral signature is strong and the interference of water vapour is relatively low (McMillan et al., 2005).

Figure 5 shows the evolution of the total column CO concentration (Texeira, 2013) every two days during the period of 11 August to 29 August 2017. The front of the CO plume reached Europe since 19 August. We find that the spatial distribution and temporal evolution of CO are strongly co-related with the UVAI. This correlation is much evident before 21 August. After

21 August, the correlation became weaker, for the UVAI in North America was decreasing fast while the CO concentration remained almost unchanged or decreased much slower, possibly due to the longer lifetime of CO. Combing the MODIS image, UVAI and CO distribution and evolution, as well as other relevant information, we conclude the aerosol plumes transported from Canada to Europe are smoke. But the link between the UTLS layer observed by Lidar and transported smoke is to be revealed.

## 2.6   CALIPSO measurements

CALIPSO measurements provide a good opportunity to investigate the vertical structure of the plumes and trace back the transport of the plumes. CALIPSO measures the backscattered signal at 532 and 1064 nm. One parallel channel and one perpendicular channel are coupled to derive particle linear depolarization ratio at 532 nm. Figure 6(a)–(f) present the profiles of the backscatter coefficient and particle linear depolarization ratio at 532 nm, corresponding to the six locations **a–f** in Figure

3. These data were obtained from the NASA Langley Research Center Atmospheric Science Data Center. The six locations are intendedly selected, falling in the region with elevated UVAI and CO concentration and following the transport pathway of the plume (in Figure 4 and 5) from Canada to Europe. Figure 6 shows the enhancements of backscatter in the UTLS. Aerosol and cloud are both possible causes of the backscatter enhancements and can be distinguished by using the particle depolarization





ratio. We have examined the temperature profiles over several sites in North America in August 2017 and found that, above 10 km, the temperature drops below -38°C, at which temperature the cloud droplets mostly turn to ice phase. The typical depolarization ratio for mixed-phase cloud is 26–35% and 36 – 46% for ice cloud (Sassen, 1995; Sivakumar et al., 2003).

Figure 6(a) and (b) show the aerosol layers observed on 14 and 15 August over the north of Canada, both locations lay in the area where MODIS observed an smoke plume on 15 August (Figure 3) and the area with high UVAI and CO concentration. The particle linear depolarization ratio is about 5% in Figure 6(a) and 10% in (b), meaning that it is an aerosol layer instead of ice or mixed-phase cloud. Figure 6(c) and (f) show UTLS layers detected at 10–20 km height, with the depolarization varying from 10% to 18%. The increasing trend of the depolarization ratio is probably due to aerosol aging. The lower layer at about 9 km in Figure 6(d) has depolarization ratio between 20% and 45% (median 32%), which falls into the category of ice or mixed-phase clouds. Profiles in Figure 6(f) were captured over Berlin at 0129 UTC on 23 August. About 150 km in the southwest, a Lidar in Leipzig measured UTLS smoke layers (Haarig et al., 2018). The particle depolarization ratio of CALIPSO at 532 nm on 23 August is consistent with ground based Lidar measurements in Lille and Leipzig, which will be presented in Section 4.

Taking account all information above, we conclude that the UTLS layers observed by the Lidar systems in northern France came from Canada by UTLS advection and underwent 10–20 days of ageing before arriving at the observation sites.

## 3   Methodology

### 3.1   Lidar data processing

#### 3.1.1   Optical depth and Lidar ratio

Raman Lidar technique (Ansmann et al., 1992) allows an independent calculation of extinction and backscatter coefficients. When the nitrogen Raman signal is not available, Klett method (Klett, 1985) is used to calculate the extinction and backscatter coefficient, based on an assumption of aerosol Lidar ratio. In this study, the UTLS aerosol layers are above 10 km where the signal-to-noise ratio of Raman channels is not sufficient to obtain high quality extinction profile of the UTLS layer, therefore, we choose Klett method. To reduce the dependence of Klett inversion on the assumption of Lidar ratio, we use a pre-calculated optical depth of the UTLS aerosol layer as an additional constraint. We test a series of Lidar ratios in the range of 10–120 sr, and apply independent Klett inversions with each assumed Lidar ratio at a step of 0.5 sr. The integral of the extinction coefficient over the UTLS layer, expressed below, is compared with the pre-calculated optical depth.

$$\tau^i(\lambda) = \int_{r_{base}}^{r_{top}} \alpha_a(\lambda, r) dr \tag{2}$$

where $\tau^i$ is the integral of extinction coefficient $\alpha_a$, derived from Klett inversion. $r$ is the distance, the subscripts *top* and *base* represent the top and and base of the UTLS aerosol layer. $\lambda$ is the Lidar wavelength. This pre-calculated optical depth is derived from the elastic channel at 355 and 532 nm. The method is widely used in cirrus clouds studies (Platt, 1973; Young, 1995). Assuming that there is only molecular scattering below and above the cirrus clouds, we can calculate the optical depth of the





cirrus clouds as below:

$$\tau^u(\lambda) = \frac{1}{2} \, ln \frac{\overline{P}_{base}(\lambda) \, r_{base}^2 \, \beta_m(\lambda, r_{top})}{\overline{P}_{top}(\lambda) \, r_{top}^2 \, \beta_m(\lambda, r_{base})} - \int\limits_{r_{base}}^{r_{top}} \alpha_m(\lambda, r) dr \tag{3}$$

where $\tau^u$ is the optical depth of the UTLS aerosol layers. $\overline{P}_{top}$ and $\overline{P}_{base}$ represent the mean Lidar signal at the top and the base of the UTLS layer. $\alpha_m$ and $\beta_m$ are the molecular extinction an backscatter coefficient. We use this method to estimate the optical depth of the UTLS layer for LILAS and IPRAL measurements. The Lidar ratio leading to the best fit of $\tau^i$ and $\tau^u$ is accepted as the retrieved Lidar ratio of the UTLS aerosol layer. We apply Klett inversion only to the UTLS aerosol layer, therefore, the impact of tropospheric aerosols is excluded.

We calculate the signal mean within a window of 0.5 km at the top and the base of the aerosol layer, to get $\overline{P}(r_{top}, \lambda)$ and $\overline{P}(r_{base}, \lambda)$. However, this method is not applicable to the MAMS Lidar measurements due to the insufficient signal-to-noise ratio above the UTLS plume. For the MAMS Lidar measurements, the columnar AOD measured by PLASMA sun photometer is used as a constraint. Klett inversion is performed to the Lidar profile from the surface to the top of the UTLS layer, assuming a vertically independent Lidar ratio.

The errors in the Lidar signal at the top and the base of the UTLS layers are considered as the major error sources of the optical depth. We estimate the error of the Lidar signal $\overline{P}(\lambda, r_{top})$ and $\overline{P}(\lambda, r_{base})$ to be 3–5%, based on the statistical error of photon distributions. According to Equation 3, the error of the optical depth, $\frac{\Delta\tau^u}{\tau^u}$, is written as:

$$\left(\frac{\Delta\tau^u}{\tau^u}\right)^2 = F_{\overline{P}_{top}} \left(\frac{\Delta\overline{P}(\lambda, r_{top})}{\overline{P}(\lambda, r_{top})}\right)^2 + F_{\overline{P}_{base}} \left(\frac{\Delta\overline{P}(\lambda, r_{base})}{\overline{P}(\lambda, r_{base})}\right)^2 \tag{4}$$

$$F_{\overline{P}_{top,base}} = \left(\frac{\overline{P}(\lambda, r_{top,base})}{\tau^u} \frac{\partial\tau^u}{\partial\overline{P}(\lambda, r_{top,base})}\right)^2 \tag{5}$$

where $\Delta$ represents the absolute error of the quantity behind it.

The error of optical depth propagates into the error of Lidar ratio and vertically integrated backscatter coefficient. Additionally, the error of the Lidar ratio also relies on the step length of Lidar ratio between two consecutive iterations and the fitting error of the optical depth of the UTLS aerosol layer, which can be limited by narrowing the step of the iteration. In our calculation, we use a step of 0.5 sr and achieve the fitting error of optical depth less than 1% which is negligible compared to the contribution of the error of optical depth. However, we can basically estimate the error of the integral of the backscatter coefficient within the UTLS aerosol layer, not the error of the backscatter coefficient profile.

### 3.1.2 Particle linear depolarization ratio

The particle linear depolarization ratio, $\delta_p$, is written as:

$$\delta_p = \frac{R\delta_v(\delta_m + 1) - \delta_m(\delta_v + 1)}{R(\delta_m + 1) - (\delta_v + 1)} \tag{6}$$





where $R$ is the backscatter ratio, $\delta_v$ is the volume linear depolarization ratio and $\delta_m$ is the molecular depolarization ratio. $R$ is defined as the ratio of the total backscatter coefficient to the molecular backscatter coefficient. $\delta_v$ is the ratio of the perpendicularly scattered signal to the parallel scattered signal, taking into account a calibration coefficient. The depolarization calibration is designated to calibrate the electro-optical ratio between the perpendicular and parallel channel and is performed following

the procedure proposed by Freudenthaler et al. (2009).

According to Equation 6, the error of particle depolarization ratio lies in three terms: the backscatter ratio $R$, volume depolarization $\delta_v$ and molecular depolarization $\delta_m$.

$$\left(\frac{\Delta\delta_p}{\delta_p}\right)^2 = F_R\left(\frac{\Delta R}{R}\right)^2 + F_{\delta_v}\left(\frac{\Delta\delta_v}{\delta_v}\right)^2 + F_{\delta_m}\left(\frac{\Delta\delta_m}{\delta_m}\right)^2 \tag{7}$$

$$F_X = \left(\frac{X}{\delta_p}\frac{\partial\delta_p}{\partial X}\right)^2, X = R, \delta_v, \delta_m \tag{8}$$

As the backscatter ratio and the volume depolarization increase, the dependence of particle depolarization ratio on the backscat-

ter ratio gets much weaker. In our study, either the depolarization ratio (at 355 nm) or the backscatter ratio (at 1064 nm), or even both, are high enough, so it allows us to conservatively assume a preliminary error level for the backscatter ratio $R$. We simply assume 20% error in the backscatter ratio. The potential error sources of the volume depolarization come from the optics and the polarization calibration. The optics have been carefully optmized and adjusted to minimize the depolarization contamination. After long-term Lidar operation and monitoring of the depolarization calibration, we conservatively expect 10% relative

errors in the volume depolarization ratio. The theoretical molecular depolarization ratio is calculated to be 0.4% with negligible wavelength dependence (Behrendt et al., 2002). In the historical record since 2013, LILAS measured molecular depolarization ratios approximately 0.8–1.3% at 532 nm channel, 1.2–1.8% at 355 nm channel and 0.7–1.0% at 1064 nm channel. IPRAL measured molecular depolarization ratio about 2.0% at 355 nm in this study. Molecular depolarization ratios measured by both LILAS and IPRAL system exceed the theoretical value. Regardless of the error in the polarization calibration, the error of

molecular depolarization ratio rises mainly from the optics, precisely, the cross-talks between the two polarization channels. The imperfections of the optics cannot be avoided, but a careful characterization is helpful to eliminate the cross-talks as much as possible (Freudenthaler, 2016). In our study, we simply assume 200% and 300% for the error of molecular depolarization ratio measured by LILAS and IPRAL system, respectively. In the following section, the total error of particle depolarization ratio is calculated according to Equation 7.

## 3.2 Aerosol inversion and radiative forcing estimation

The $3\beta + 2\alpha$ from Lidar observations can be inverted to obtain particle microphysical parameters. The regularization algorithm is used to retrieve size distribution, wavelength-independent complex refractive indices, particle number, surface and volume concentrations (Müller et al., 1999; Veselovskii et al., 2002). DRF estimation is performed with the retrieved aerosol microphysical properties, the vertical profile of the UTLS plume and surface BRDF (Bidirectional Reflectance Distribution Function) parameters from AERONET. In this study, we apply the forward model of GRASP (Generalized Retrieval of Aerosol and Surface Properties) to calculate the DRF effect of the UTLS aerosol layer. GRASP is the first unified algorithm developed for





characterising atmospheric properties gathered from a variety of remote sensing observations. GRASP is a highly versatile algorithm. Depending on the input data, GRASP can retrieve columnar and vertical aerosol properties and surface reflectance (Dubovik et al., 2014). As a branch of GRASP algorithm, GARRLiC (Generalized Aerosol Retrieval from Radiometer and Lidar Combined data) algorithm was developed for the inversion of coincident single or multi-spectral Lidar and sun photometer

measurements(Lopatin et al., 2013; Bovchaliuk et al., 2016). The two main modules of GARRLiC/GRASP are the forward model and the numerical inversion module. The forward module simulates the atmospheric radiation by using radiative transfer and by accounting for the interaction between light and trace gases, aerosols and underlying surfaces. The numerical inversion module follows the multi-term least squares method strategy and derives several groups of unknown parameters that fit the observations. Therefore, we can use the forward model of GARRLiC/GRASP to estimate the radiative effect of the UTLS

aerosols.

## 4    Results and analysis

### 4.1    Overview of retrieved optical parameters

We selected and averaged the Lidar measurement in 10 time intervals, among which five intervals are from LILAS system in Lille: 2200 (24 August) $-0030$ UTC (25 August), $1300-1600$ UTC, $1600-1800$ UTC (29 August), $2000-2300$ (31 August)

and 2300 (31 August)$-0200$ UTC (01 September); two intervals from IPRAL system in Palaiseau: $1600-1800$ UTC and $1920-2120$ UTC (28 August) and three intervals from the mobile Lidar in MAMS system (29 August): $1400-1500$ UTC (corresponding spatially to 100 km distance from Palaiseau to Compiègne), $1500-1545$ UTC (100 km on the route from Compiègne to Arras) and $1615-1630$ UTC at Lille.

Figure 7 shows the optical depth of the UTLS layer varying from 0.05 to 0.23 (at 532 nm). The spectral dependence of the

optical depth of 355 nm and 532 nm is very weak. The maximal optical depth of the UTLS layer was observed in the afternoon of 29 August, between 1600 and 1800 UTC. LILAS system observed aerosol optical depth of $0.20 \pm 0.04$ at 355 nm and $0.21 \pm 0.04$ at 532 nm. As discussed in Section 2.1, the columnar AOD at 532 nm from AERONET increased by about 0.20 after the presence of the UTLS layer, which agrees well with the derived optical depth of the UTLS layer. The minimum of the optical depth appeared in the night of 31 August 2017, giving $0.04 \pm 0.02$ at 355 nm and $0.05 \pm 0.02$ at 532 nm. The optical

depth of the UTLS layer along the route, observed by MAMS, are as follows: 0.19 over a distance of 100 km North from Palaiseau, 0.23 along 100 km of the middle of the transect from Compiègne to Arras and 0.22 when arriving at Lille. The error of optical depth derived from the mobile Lidar system is not estimated, because of inversion of the Lidar in MAMS follows a different procedure and includes the impact of tropospheric aerosols, as a result, the error is difficult to quantify. Here we present the optical depth estimated from mobile Lidar for comparison.

Table 1 summarizes the Lidar ratio and particle depolarization ratio in the UTLS aerosol layer. Lidar ratios vary between $54 \pm 9$ sr and $58 \pm 23$ sr at 532 nm and between $31 \pm 15$ sr and $45 \pm 9$ sr at 355 nm. The results from two different Lidar systems and with different observation time agree well, indicating that the properties of the UTLS layer are spatially and temporally stable. We derived higher Lidar ratio at 532 nm than at 355 nm which is a characteristic feature of aged smoke and has been



observed in previous studies (Wandinger et al., 2002; Murayama et al., 2004; Müller et al., 2005; Sugimoto et al., 2010). In the night of 31 August, the error of Lidar ratio is about $30 - 35\%$, relatively higher than the other days because of the low optical depth. Although the error varies, the mean values of derived Lidar ratio are relatively stable. The particle depolarization ratio decreases as wavelength increases. At 1064 nm channel, the particle linear depolarization ratio is very stable, varying

from $4.0 \pm 0.6\%$ from $5.0 \pm 0.8\%$. At 532 nm channel, the depolarization is also stable, varying from $18 \pm 3\%$ to $20 \pm 3\%$. The particle linear depolarization ratio at 355 nm increased from $23 \pm 3\%$ on 24 August to $28 \pm 4\%$ on 31 August. However, the increase is merely within the range of the uncertainties, thus making it difficult to conclude. The particle depolarization ratio at 532 nm is in good agreement with CALIPSO observations shown in Figure 6(c)–(f). The particle depolarization ratio at 355 nm measured by LILAS and IPRAL system is also consistent, meaning that the UTLS aerosol layer observed in Lille and

Palaiseau are likely originated from the same source. A Leipzig Lidar also observed UTLS aerosol layers on 22 August 2017. They measured 23.4% at 355 nm, 18.4% at 532 nm and 4.2% at 1064 nm (Haarig et al., 2018), which are in good agreements with our observations.

The errors of particle depolarization ratio are calculated with the method in Section 3. Considering the measurements in the night of 31 August, the optical depth is the lowest and the error is expected to be higher than the other cases. The backscatter

ratio, volume depolarization ratio and volume depolarization ratio at 355 nm are approximately: 3.5 (20%), 15% (10%) and 0.4% (200%). The values in the parentheses are the relative errors of the quantity on the left. The resulting error of particle depolarization is about 14%. At 532 nm channel, we derive 12% of error for the particle depolarization ratio when the backscatter ratio, volume depolarization ratio and molecular depolarization ratio are: 10 (20%), 15% (10%) and 0.4% (200%). In the same way, we derive less than 11% of error for the particle depolarization ratio at 1064 nm. The error at 355 nm is estimated to be

higher than 532 and 1064 nm as the interferences of molecular scattering is stronger at this channel. When the layer is optically thicker, for example, 24 August, the error of 355 nm is estimated to be less than 13%. Conservatively, we use 15% for the error of the particle depolarization ratio at three wavelengths, but the real error should be below this level.

## 4.2   Case study

### 4.2.1   Optical properties

We selected the night measurements of 24 August in Lille and 28 August in Palaiseau as two examples. The two systems were operating independently, so that the results from two different systems that measured at different time can be regarded as verifications for each other.

**24 August 2017, Lille**

Figure 8 shows the retrieved optical properties of the UTLS layer observed by LILAS system in the night of 24 August in

Lille. The UTLS aerosol layer is between 17 and 18 km, and we retrieved the extinction and backscatter profiles by assuming that the Lidar ratios are 36 sr at 355 nm and 54 sr at 532 nm. The Lidar ratio at 1064 nm channel is assumed to be 60 sr. The extinction coefficient within the layer is about $0.14 - 0.22 \text{ km}^{-1}$ at 355 nm and 532 nm. The extinction-related Ångström exponent for 355 and 532 nm is around $0.0 \pm 0.3$, the backscatter-related Ångström exponent at corresponding wavelengths is



about $1.0 \pm 0.3$. The particle depolarization ratios decrease as wavelength increases. The vertical variations of the extinction and backscatter-related Ånsgtröm exponent and particle depolarization is weak, indicating that the observed aerosol layer is homogenous.

**28 August 2017, Palaiseau**

Figure 9 shows the retrieved optical parameters from IPRAL observations at $1920 - 2120$ UTC, 28 August 2017 in Palaiseau. The thickness of the UTLS layer is about 2.5 km, spreading from 17 km to 19.5 km. Klett inversion was applied with estimated Lidar ratio of 36 sr at 355 nm and 58 sr at 532 nm. At 1064 nm the Lidar ratio was assumed to be 60 sr. The maximum extinction coefficient in the layer reached $0.12$ km$^{-1}$ at 532 nm. The extinction-related Ångström exponent between 355 nm and 532 nm is about $-0.06 \pm 0.3$. The corresponding backscatter Ångström exponent is about $1.2 \pm 0.3$. The particle linear depolarization ratio at 355 nm is about $27 \pm 4\%$. The particle linear depolarization ratio at 355 nm, extinction and backscatter-related Ångström exponent between 355 nm and 532 nm do not show evident vertical variations.

The back trajectories (Stein et al., 2015; Rolph et al., 2017) in Figure 10 show that the UTLS aerosol layers observed by LILAS on 24 August and IPRAL on 28 August were transported by UTLS advection from eastern Canada, near the coast of the Atlantic Ocean. The UVAI map on 19 August (in Figure 4(e)) shows that the smoke plume was leaving the Canadian territory and spread over large areas in the Labrador Sea. Besides, CALIPSO profile on 19 August, i.e. Figure 6(d) shows an elevated aerosol layer at $16 - 18$ km near this region.

### 4.2.2 Microphysical properties

Regularization algorithm is applied to the vertically averaged extinction coefficients (at 355 and 532 nm) and backscatter coefficients (at 355, 532 and 1064 nm) in Figure 8 and Figure 9. Treating non-spherical particles is a challenging task. Many studies have been done to model the light scattering of non-spherical particles. The spheroid model was used to retrieved dust properties (Dubovik et al., 2006; Mishchenko et al., 1997; Veselovskii et al., 2010). But it is not clear if this model is applicable to soot particles with complicated morphology. The size of smoke particles is expected not too big so that we choose to apply regluarization algorithm with sphere model. The particle linear depolarization ratio is not used in the retrieval, and the spectral dependence of complex refractive indices is also ignored in the retrieval. The derived effective radius ($R_{eff}$), volume concentration ($V_c$), the real ($m_R$) and imaginary ($m_I$) part of the refractive indices are summarized in Table 2. The errors of the retrieved parameters have been discussed in the relevant papers (Müller et al., 1999; Veselovskii et al., 2002; Pérez-Ramírez et al., 2013).

The retrieved particle size distributes in the range of 0.1 to 1.0 $\mu$m, with effective radius (volume-weighted sphere radius) of $0.33 \pm 0.10$ for both Palaiseau data and Lille data. The volume concentration is $15 \pm 5$ $\mu$m$^{-3}$cm$^3$ for Palaiseau data and $22 \pm 7$ $\mu$m$^{-3}$cm$^3$ for Lille data. The real part of the complex refractive indices retrieved from Lille and Palaiseau data are also in good agreement, giving $1.55 \pm 0.05$ and $1.52 \pm 0.05$ for the real part, and $0.028 \pm 0.014$ and $0.021 \pm 0.010$ for the imaginary part. The derived aerosol microphysical properties from Palaiseau and Lille data are consistent.





### 4.2.3 Direct radiative forcing effect

The UTLS plumes observed on 24 and 28 August in Lille and Palaiseau are optically thick, with extinction coefficient about 10 times higher than in the volcanic ash observed by Ansmann et al. (1997) in April 1992, 10 months after the eruption of Mount Pinatubo. The radiative forcing imposed by the observed layers is a curious question. We input the retrieved microphysical properties into GARRLiC/GRASP to estimate the DRF effect of the UTLS plumes in Lille and Palaiseau. The calculation

includes the $0.2 - 4.0$ $\mu$m spectral range. We assume the vertical volume concentration of aerosols follows the extinction profile in Figure 8 and 9. The surface BRDF parameters for Lille and Palaiseau are taken from AERONET. The upward and downward flux/efficiencies, as well as the net DRF ($\Delta F$, with respect to a pure Rayleigh atmosphere) of the UTLS aerosol layers are calculated and Table 3 shows the daily averaged net DRF (W/m$^2$) at four levels: at the bottom of the atmosphere (BOA), below the UTLS layer, above the UTLS layer and at the top of the atmosphere (TOA). For the layer observed in Lille

on 24 August, the top and base of the UTLS are selected as: 18.4 km and 16.7 km and for Palaiseau observations, they are 20 km and 17.0 km.

At the top of the atmosphere, the net DRF flux is estimated to be -1.2 Wm$^{-2}$ and -3.5 Wm$^{-2}$ for Lille and Palaiseau data, respectively. The corresponding forcing efficiencies are -7.9 Wm$^{-2}\tau^{-1}$ and -21.5 Wm$^{-2}\tau^{-1}$. At the bottom of the atmosphere, the net DRF flux is estimated to -12.3 Wm$^{-2}$ for Lille data and -14.5 Wm$^{-2}$ for Palaiseau data. The corresponding forcing

efficiencies are -79.6 Wm$^{-2}\tau^{-1}$ and -89.6 Wm$^{-2}\tau^{-1}$. We noticed that the difference in net DRF flux between the layer top and layer base is significant. For Lille data, we obtained 9.9 W/m$^2$ of difference between the top and the base of the UTLS layer and for Palaiseau, we obtained 11.1 W/m$^{-2}$. Because of the high imaginary part of refractive indices, the UTLS aerosols have the capacity of absorbing the incoming radiation, thus reducing the upward radiation at the top of the UTLS layer and the downward radiation at the base of the UTLS aerosol layer. The heating rate of the UTLS layer is estimated to be 3.3 K/day for

Palaiseau data and 3.7 K/day for Lille data. This qualitatively explains the increase of temperature within the UTLS layer, as observed by the radiosonde measurements shown in Figure 2.

## 5 Discussion

Based on the satellite measurements of UVAI, CO concentration and CALIPSO profiles, we can conclude that the UTLS layers observed over northern France are originated from Canadian smoke. The layers are transported by UTLS advection and are

characterized by high UVAI, CO concentration, together with enhanced depolarization ratios. The smoke particles were lofted to the UTLS probably by the pyro-cumulus clouds observed by GEOS-15 on 12 August 2017. The UTLS smoke plumes reported in this study have traveled for more than 10 days from the source region, considering that the smoke plumes were lofted on 12 August and then observed by the Lidar systems during 24–31 August. The high altitude of the smoke plumes prevented them from mixing with many other aerosol types during the transport. So the observed plumes are most likely to be aged smoke

particles.

The measurements revealed high depolarization ratios in the UTLS aerosol at 355 and 532 nm. In particular, the range of depolarization at 355 nm is $23 \pm 3\%$ to $28 \pm 4\%$, while at 532 nm it is about $19 \pm 3\%$. The depolarization ratio at 1064 nm is





significantly lower, about $4.5 \pm 0.8\%$. Similar spectral dependence of depolarization ratio: 24%, 9% and 2% at 355, 532 and 1064 nm, respectively, was observed by Burton et al. (2015) in the smoke plume from North American wildfires. Depolarization ratio of 7% and 1.9% at 532 and 1064 nm, respectively, were observed in the smoke plume from southwest Canadian fires (Burton et al., 2012). In both cases, the distance between the emission and the observation sites was smaller than in our case. The increase of particle depolarization of smoke layers versus transport time has been indicated by CALIPSO data. The

processes of the light scattering, leading to high depolarization ratios of smoke particles are not well understood. In previous studies, smoke mixed with soil particles was suggested to be the explanation (Fiebig et al., 2002; Murayama et al., 2004; Müller et al., 2007a; Sugimoto et al., 2010; Burton et al., 2012; Haarig et al., 2018). Strong convections occurred in fire activities, in principle are capable to lift soil particles into the smoke plume (Sugimoto et al., 2010). However, Murayama et al. (2004) suggested that the coagulation of smoke particles to the clusters with complicated morphology is a more reasonable explanation

because they found no signature of mineral dust after analyzing the chemical compositions of the smoke samples. Mishchenko et al. (2016) modeled the spectral depolarization ratios observed by Burton et al. (2015) and found that such behavior is resulted from complicated morphology of smoke particles.

Figure 11 shows the depolarization ratios at 532 nm as a function of transport time, using the observations from the present study and previous the publications (Fiebig et al., 2002; Fromm et al., 2008; Sugimoto et al., 2010; Burton et al., 2012; Dahlköt-

ter et al., 2014; Haarig et al., 2018). Depolarization ratio increases with the transport time, and the increase is more significant in the first 10 days. Müller et al. (2007b) found that the size of free-tropospheric particles which originated from boreal forest fires grows with transport time in the same manner as in Figure 11. The observed behavior of depolarization ratio can be the indication of the growth of smoke particle aggregates during the transportation. Kahnert et al. (2012) modeled the optical properties of light absorbing carbon aggregates (LAC) embedded in a sulfate shell. It was found that the particle depolarization

ratio increases with the aggregate radius (volume-equivalent sphere radius). For the case of 0.4 $\mu$m aggregate radius and 20% LAC volume fraction, the computed depolarization ratios are 12–20% at 304.0 nm, 8.0–18% at 533.1 nm and about 1.5% at 1010.1 nm, which are comparable with our results.

High depolarization ratio with similar spectral dependence has been observed for fine dust particles in laboratory measurements. Miffre et al. (2016) measured the depolarization ratio of two Arizona Test Dust samples at backscattering angle. The

radii of the dust samples distributed within 1 $\mu$m. They obtained higher depolarization ratio at 355 nm than at 532 nm and the depolarization ratio at both wavelengths are over 30%. In the study of Järvinen et al. (2016), over 200 dust samples were used to measure the near-backscattering (178°) properties and it is found that, for fine-mode dust, the depolarization has a strong size dependence. Järvinen et al. (2016) obtained about 12–20% and 25–30% depolarization ratio for corresponding particle size parameters at 355 and 532 nm in this study. Sakai et al. (2010) measured the depolarization of Asian and Saharan dust

in the backscattering direction and obtained 14–17% at 532 nm for the samples with only submicrometer particles and 39% for the samples with high concentration of supermicrometer particles. In this study, soils particles are likely to have partial contribution to the depolarization ratio, however, the presence of soil particles is not sufficient to explain the increase of depolarization ratio versus transport time, for which the coagulation of smoke particles should be a more realistic explanation.

The derived Lidar ratios are in the range of $31 \pm 15$ sr to $45 \pm 9$ sr for 355 nm and $54 \pm 12$ sr to $58 \pm 9$ sr for 532 nm. Consid-



ering the uncertainties of the Lidar ratio, the derived values and the spectral dependence agree well with previous publications (Müller et al., 2005; Sugimoto et al., 2010; Haarig et al., 2018) for aged smoke observations. The retrieved effective radius is about $0.33 \pm 0.10$ $\mu$m, which is larger than the values of fresh smoke observed near the source region of biomass burning (O'Neill et al., 2002; Nicolae et al., 2013). The increase of the smoke particle size during transport in the free-troposphere was previously reported in Müller et al. (2007b). In particular, the retrieved particle size agrees well with the observed smoke

aerosol transported from Canada to Europe (Wandinger et al., 2002; Müller et al., 2005). Müller et al. (2007b) found that the effective radius increased from $0.15 - 0.25$ $\mu$m ($2 - 4$ days after the emission) to $0.3 - 0.4$ $\mu$m after $10 - 20$ days of transport time, which is consistent with our results. Colarco et al. (2004) assessed that the coagulation process during the transport could explain the growth of particle size and the decrease of extinction-related Ångström exponent.

The real part of the refractive indices obtained in this study is 1.52±0.05 for Palaiseau data and 1.55±0.05 for Lille data,

without considering the spectral dependence. The values are consistent with the results for smoke aerosol in the troposphere (Dubovik et al., 2002; Wandinger et al., 2002; Taubman et al., 2004; Müller et al., 2005). As to the imaginary part, we derived $0.021 \pm 0.010$ from Palaiseau data and $0.028 \pm 0.014$ from Lille data. Typical values of the imaginary part for smoke aerosols derived in AERONET are about $0.015 \pm 0.004$, which is consistent with our results. The relative humidity in the aerosol layer has an impact on the refractive indices. Smoke in dry conditions have higher refractive indices than that in wet condition.

In some studies, the relative humidity is not mentioned and it makes the results difficult to compare. The contamination of other aerosol types during the transport is also a potential cause of the modification of aerosol properties, and its impact is not limited to the refractive indices. In this study, the smoke layers we observed were lofted to the UTLS in the source region and then transported to the observation sites by advection. It is isolated from other tropospheric aerosol sources and not likely to mix with them during the transport. The relative humidity in the UTLS layer is below 10%, according to the radiosonde

measurements. Our study provides a reference for aged smoke aerosols in a dry condition.

The retrieved particle parameters allow an estimation of direct aerosol radiative forcing. We derived -79.6 Wm$^{-2}\tau^{-1}$ and -7.9 Wm$^{-2}\tau^{-1}$ for the DRF efficiencies at the bottom and the top of the atmosphere for Lille data. And for Palaiseau data, we derived -89.6 Wm$^{-2}\tau^{-1}$ and -21.5 Wm$^{-2}\tau^{-1}$. It indicates that the observed UTLS aerosol layers reduce strongly the radiation reaching the terrestrial surface mainly by absorbing the solar radiation. Derimian et al. (2016) evaluated the radiative

effect of several aerosol models, among which the daily net DRF efficiency of biomass burning aerosols is estimated to be -74 Wm$^{-2}\tau^{-1}$ to -54 Wm$^{-2}\tau^{-1}$ at the bottom of the atmosphere. Mallet et al. (2008) studied the radiative forcing of smoke and dust mixture over Djougou and derived -68 Wm$^{-2}\tau^{-1}$ to -50 Wm$^{-2}\tau^{-1}$ for the DRF efficiency at the bottom of the atmosphere. The uncertainties of the DRF estimation could be large, because the uncertainty of the retrieved microphysical properties is high. Our results show stronger forcing efficiencies, but are comparable with the values in the publications. Addi-

tionally, the mean heating rate of the UTLS aerosol layer is estimated to be 3.7 K/day for Lille data and 3.3 K/day for Palaiseau data, which qualitatively supports the temperature increase within the UTLS smoke layer. The warming effect in the layer is potentially responsible for the upward movements of soot-containing aerosol plumes observed by Laat et al. (2012).



## 6  Conclusion

In 2017 Summer, large-scale wildfires spread in the west and north of Canada. In the mid-August, severe fire activities generated strong convections that lofted smoke plumes up to the UTLS. The smoke plumes were transported by advection in the UTLS and observed by the Lidar systems in northern France after long-range transport. Multiple satellite sensors recorded the long-range transport of the smoke plumes. The smoke plumes have undergone more than 10 days of aging process before being observed by the Lidar systems in northern France. The optical and microphysical properties derived from Lidar observations revealed important features of aged smoke, including the optical depth, Ångström exponent, Lidar ratio, depolarization ratio, refractive indices, particle size and concentrations. The increase of smoke particle depolarization ratios versus transport time is first presented and points to smoke particle coagulation during the transport. The DRF estimation indicated that the UTLS aerosols strongly reduce the radiation reaching the bottom of the atmosphere by absorption. The derived heating rate of the UTLS layer agreed with the temperature increase within the layer, according to the radiosonde measurements. This study provides a good reference for the characterization of long-transported smoke aerosols and shows the capability of multi-wavelength Raman Lidar in aerosol profiling and characterization.

*Data availability.*  All the data used in this paper are available upon request to the corresponding author.

*Acknowledgements.*  We wish to thank ESA/IDEAS program who supported this work. FEDER/Region Hauts-de-France and CaPPA Labex are acknowledged for their support for LILAS multiwavelength Raman LiDAR and MAMS system. H2020-ACTRIS-2/LiCAL Calibration center, ACTRIS-France, ANRT France, CIMEL Electronique and Service National Observation PHOTONS/AERONET from CNRS-INSU are acknowledged for their support. The development of Lidar retrieval algorithms was supported by Russian Science Foundation (project 16-17-10241). The authors would like to acknowledge the use of GRASP inversion algorithm (http://www.grasp-open.com) in this work. The SIRTA observatory and supporting institutes are acknowledged for providing IPRAL data. Finally we thank all the co-authors for their kind cooperation and professional help.



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





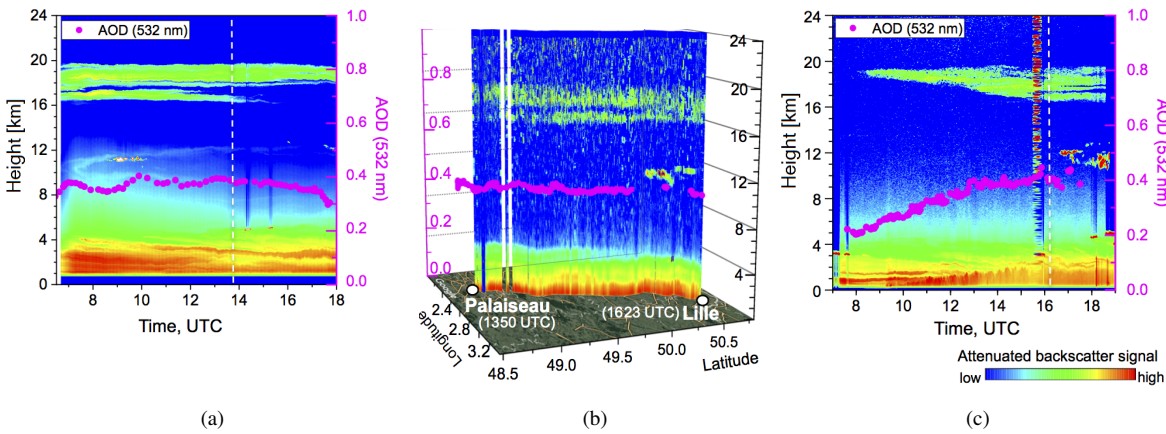

**Figure 1.** Lidar range-corrected signal and columnar AOD from sun photometer at 532 nm, 29 August 2017. (a) IPRAL system in Palaiseau. (b) MAMS Lidar on-route from Palaiseau to Lille. (c) LILAS in Lille. Columnar AOD measurements are interpolated from AERONET (Lille and Palaiseau) and PLASMA (mobile system) measurements. MAMS started from Palaiseau at 1353 UTC and arrived in Lille at 1623 UTC. The departure and arriving time are indicated in (a) and (c) with the white dashed lines.

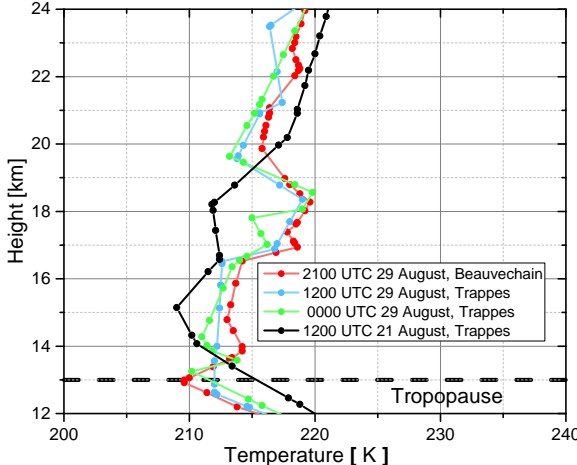

**Figure 2.** Temperature profiles from the radiosonde measurements. The green and cerulean lines are the temperature profiles of Trappes at 0000 and 1200 UTC, 29 August 2017. The red line shows the Beauvechain data at 2100 UTC 29 August 2017. The black line is for 1200 UTC, 21 August, Trappes. The horizontal black dashed line at 13 km represents the approximate position of the tropopause.





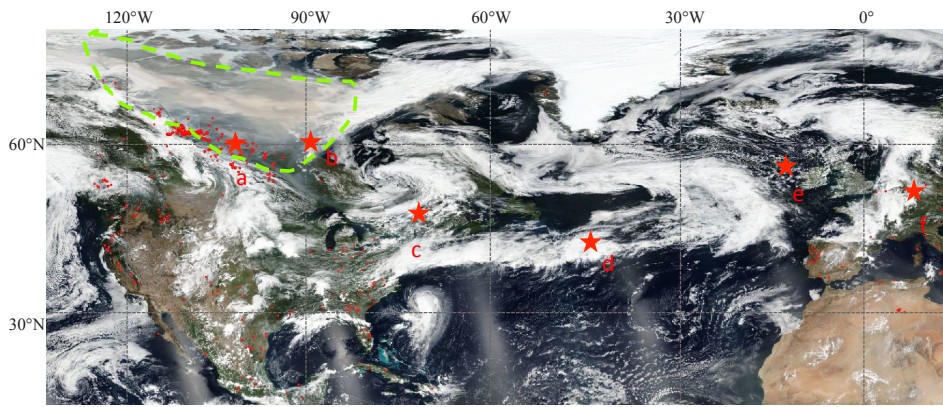

**Figure 3.** The corrected surface reflectance overlaid with fire and thermal anomalies from MODIS (15 August 2017). The region marked with green dashed line in the northwest indicated a plume generated by fire activities. Six locations (labeled as red stars) on the tracks of CALIPSO are selected: **a** (61.47°N, 106.44°W), **b** (62.79°N, 91.54°W), **c** (46.97°N, 72.22°W), **d** (42.27°N, 42.08°W), **e** (55.97°N, 12.54°W) and **f** (52.37°N, 13.47°E). The corresponding overpass date is 14, 15, 17, 19, 21 and 23 August 2017.







**Figure 4.** OMPS NM daily UVAI products during 11 to 29 August 2017. The results are plotted every two days. Grey colour indicates areas with no retrievals.







**Figure 5.** Total CO concentration (molecules/cm$^2$) retrieved from AIRS. The maps are plotted every two days during 11 and 29 August 2017.





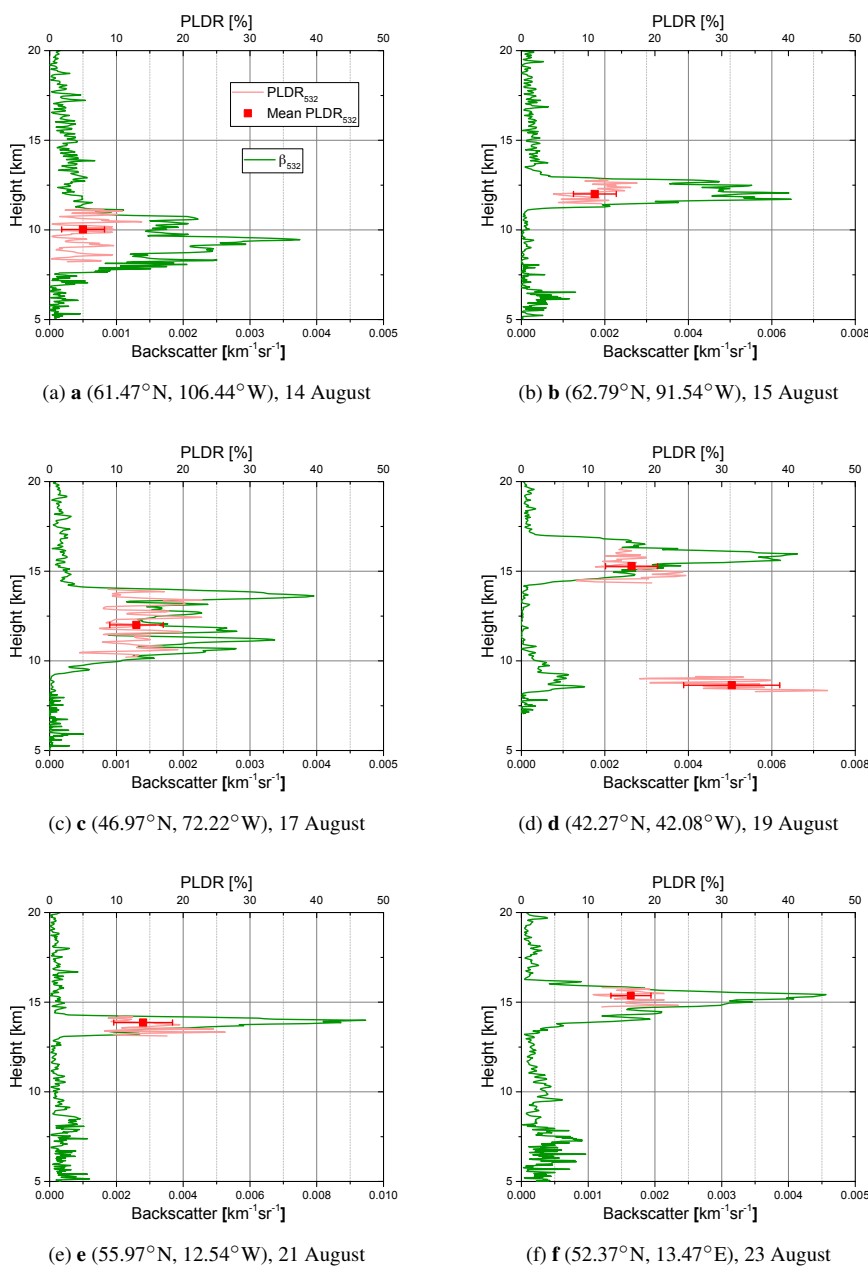

(a) **a** (61.47°N, 106.44°W), 14 August

(b) **b** (62.79°N, 91.54°W), 15 August

(c) **c** (46.97°N, 72.22°W), 17 August

(d) **d** (42.27°N, 42.08°W), 19 August

(e) **e** (55.97°N, 12.54°W), 21 August

(f) **f** (52.37°N, 13.47°E), 23 August

**Figure 6.** The profiles of backscatter coefficient and particle linear depolarization ratio (PLDR). Figure (a)-(f) correspond to the six locations **a** – **f** in Figure 3. The corresponding CALIPSO tracks are (a) 09:50:19, 14 August 2017; (b) 08:54:37, 15 August 2017; (c) 07:03:13, 17 August 2017; (d) 06:50:44, 19 August 2017; (e) 03:20:25, 21 August 2017 and (f) 01:29:01, 23 August 2017. 20 profiles are averaged over these six locations. The green and pink solid lines represent backscatter coefficient and particle linear depolarization ratio, respectively. The red squares with error bars represent the mean particle linear depolarization ratio and the standard deviation within each layer.





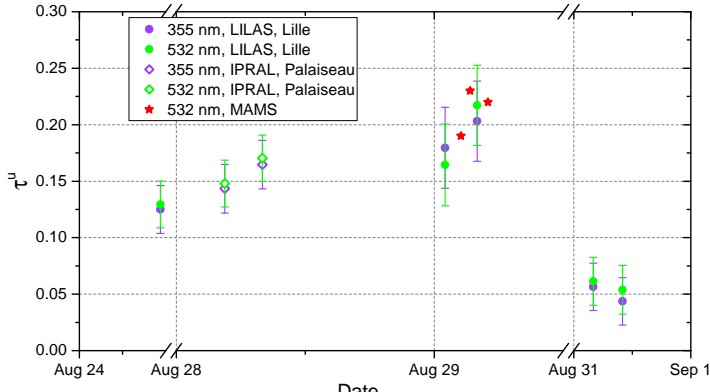

**Figure 7.** Optical depth of the UTLS layer at 355 and 532 nm estimated from Lidar signals. The optical depth estimated from LILAS (in Lille) is plotted with green (532 nm) and violet solid circles (355 nm). Optical depth calculated from IPRAL (in Palaiseau) is plotted with green (532 nm) and violet (355 nm) open diamonds. The red stars represent the optical depth calculated from the MAMS Lidar.

**Table 1.** Retrieved Lidar ratios (LR) and particle linear depolarization ratios (PLDR) from multi-wavelength Lidar systems LILAS in Lille and IPRAL in Palaiseau.

| Lidar system | LILAS, Lille | | | | | IPRAL, Palaiseau |
|---|---|---|---|---|---|---|
| Date | 24 August | 29 August | | 31 August | | 28 August |
| Time (UTC) | 2200 – 0030 | 1300 – 1600 | 1600 – 1800 | 2000 – 2300 | 2300 – 0200 | 1920 – 2120 |
| $LR_{355}$ (sr) | $35 \pm 6$ | $45 \pm 9$ | $41 \pm 7$ | $34 \pm 12$ | $31 \pm 15$ | $36 \pm 6$ |
| $LR_{532}$ (sr) | $54 \pm 9$ | $56 \pm 12$ | $54 \pm 9$ | $58 \pm 20$ | $58 \pm 23$ | $58 \pm 7$ |
| $PLDR_{355}$ (%) | $23 \pm 3$ | $24 \pm 4$ | $24 \pm 4$ | $28 \pm 4$ | $28 \pm 4$ | $27 \pm 4$ |
| $PLDR_{532}$ (%) | $20 \pm 3$ | $18 \pm 3$ | $19 \pm 3$ | $18 \pm 3$ | $18 \pm 3$ | – |
| $PLDR_{1064}$ (%) | $5.0 \pm 0.8$ | $4.0 \pm 0.6$ | $4.5 \pm 0.7$ | $4.7 \pm 0.7$ | $4.7 \pm 0.7$ | – |



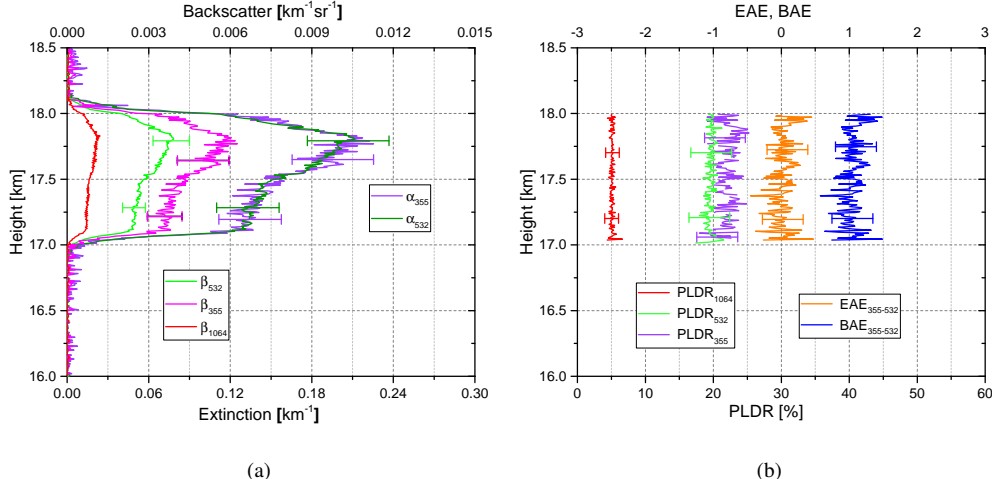

**Figure 8.** (a) Extinction and backscatter coefficient, (b) particle linear depolarization ratio (PLDR), the extinction-related Ångström exponent (EAE) and backscatter-related Ångström exponent (BAE) retrieved from LILAS observations between 2200 UTC, 24 August 2017 and 0030 UTC, 25 August 2017, Lille. The errors of extinction, backscatter coefficient and corresponding Ångström exponent at 355 and 532 nm are attributed to the error of the optical depth.

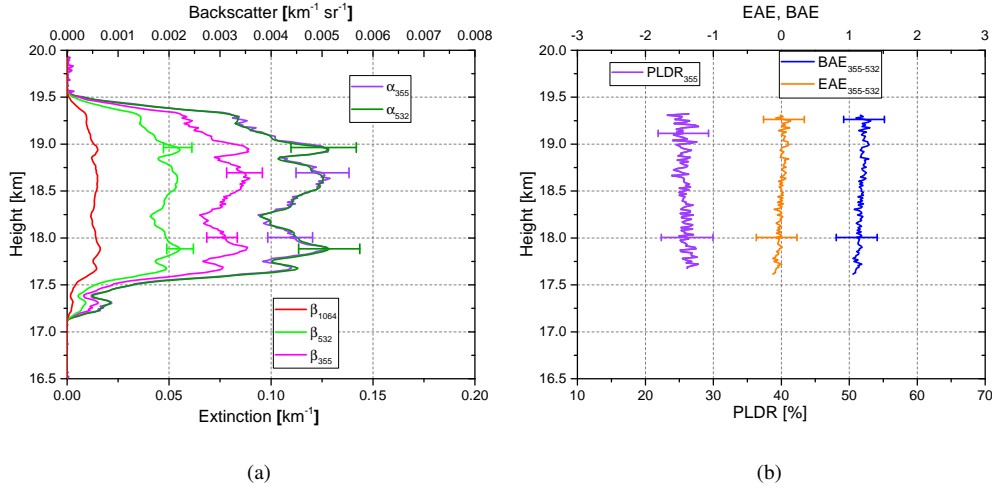

**Figure 9.** (a) Extinction and backscatter coefficient, (b) the particle linear depolarization ratio (PLDR) at 355 nm, the extinction-related Ångström exponent (EAE) and backscatter-related Ångström exponent (BAE) (between 355 nm and 532 nm) retrieved from IPRAL observations between 1920 and 2120 UTC, 28 August 2017, Palaiseau.





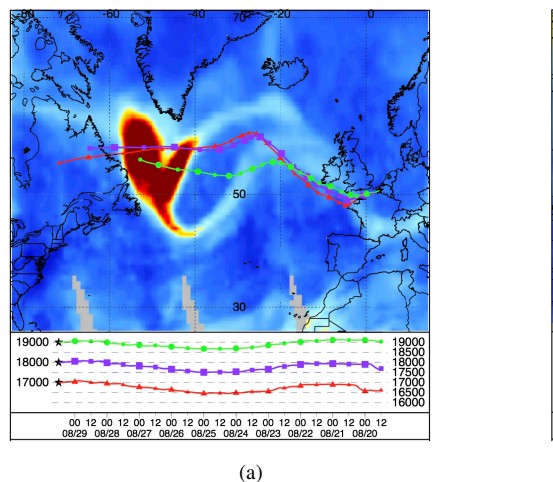
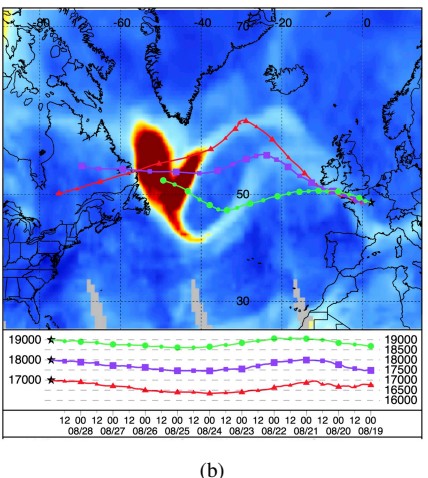

(a)                                                    (b)

**Figure 10.** The back trajectories from the HYSPLIT model overlaid with UVAI map on 19 August 2017 from OMPS NM. (a) The back trajectory of UTLS layer observed in Lille, starting time is 2200 UTC, 24 August 2017. (b) Back trajectory of the UTLS layer observed in Palaiseau, the trajectory starting time is 2200 UTC, 28 August 2017.

**Table 2.** Retrieved microphysical properties using the Lidar data in Lille and Palaiseau. Extinctin and backscatter coefficients shown in Figure 8(a) and 9(a) are averaged in the range of 17–18.0 km and 17.5–19.5 km, respectively. The averaged extinction and backscatter coefficients are used as the input of regularization algorithm to retrieve particle microphysical properties.

|  | $R_{eff}, \mu m$ | $V_c, \mu m^{-3} cm^3$ | $m_R$ | $m_I$ |
|---|---|---|---|---|
| Lille, 24 August | $0.33 \pm 0.10$ | $22 \pm 8$ | $1.55 \pm 0.05$ | $0.028 \pm 0.014$ |
| Palaiseau, 28 August | $0.33 \pm 0.10$ | $15 \pm 5$ | $1.52 \pm 0.05$ | $0.021 \pm 0.011$ |

**Table 3.** Daily averaged net DRF flux calculated by GARRLiC/GRASP. Aerosol microphysical properties in Table 2 and aerosol vertical distributions in Figure 8(a) and 9(a) are used to calculate the DRF effect at the following four vertical levels.

| $\Delta F$ (W/m$^2$) | TOA | BOA | layer top | layer base |
|---|---|---|---|---|
| Lille, 24 August | -1.2 | -12.3 | -2.1 | -12.0 |
| Palaiseau, 28 August | -3.5 | -14.5 | -2.5 | -13.6 |





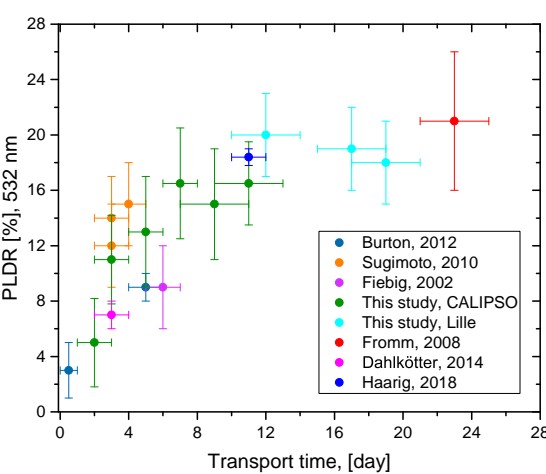

**Figure 11.** The variation of particle linear deploarization ratio at 532 nm versus transport time. The transport time in this study and in Haarig et al. (2018) is estimated by assuming that the smoke plumes were emitted on 12 August 2017. The particle depolarization ratio (red solid circle) in Fromm et al. (2008) is calculated using the volume depolarization ratio and backscattering ratio measured on 24 June, 2001. The transport time for Burton et al. (2012) was estimated from HSPLIT back trajectories (Stein et al., 2015; Rolph et al., 2017), as it was not available in the paper. The other transport times are taken from the original publication.