# Peer review of "Long-range transported Canadian smoke plumes in the lower stratosphere over northern France"

_Atmospheric Chemistry and Physics, 2018_

## Short Comment (SC1)

General remark:

The paper is well written and focusses on an interesting atmospheric topic: Long-range transport of pyrocumulonimbus-related Canadian fire smoke in the stratosphere observed over France in August 2017.

An original and unique measurement strategy is selected: Two lidars at Palaiseau and Lille and one mobile system that measured the smoke during a travel from Palaiseau to Lille.

The highlight is the retrieval of the spectrum of the particle linear depolarization ratio measured at three wavelengths. There is a strong decrease of the depolarization ratio from 25% in the UV down to less than 5% in the near IR.

However, the discussion of the results is partly confusing and must be improved.

Major revisions are necessary.

Detailed comments:

The title is misleading. The focus is on lidar observations (in France) of aerosol layers several kilometers above the tropopause, and not below the tropopause (upper troposphere). So: … smoke aerosols in the lower stratosphere …. would be correct. Furthermore, the title does not indicate where you made these observations: …. Long range transport of Canadian fires smoke towards Europe observed over northern France… , and not, for example, in eastern Asia or in the Arctic. Please improve!

P1: The abstract must be updated after the requested changes.

P2, L2: All abbreviations must be explained when they are mentioned for the first time (in the main text). The abstract is a stand-alone text, and does not count in this respect.

P2, L25. After the foregoing paragraph, a paragraph is missing in which the literature is reviewed which is already available regarding the Canadian smoke period in August 2017: Khaykin et al. (GRL, 2018), Ansmann et al. (ACP, 2018), Haarig et al. (ACP, 2018). If there are more, please mention them as well. Such a literature of foregoing work is (always) required in the Introduction. What is already done and thus known? What is our new contribution?

P3, L1…. : Again, please explain: LILAS, IPRAL, PLASMA, MAMS etc…

It is quite confusing that Section 2, that contains first observations and in-depth analysis of all the satellite observations of the smoke including CALIOP (space lidar) measurements, …. is then interrupted by a 'dry' lidar methodology section. It would be better to have the technical section first (as section 2, or as an Appendix) and then all the observations and discussions in the follow-up sections, continuously in sections 3, 4, 5…. without a break…

P3, L28: With the AOD of the smoke layer and the vertical extent of the layer (from the lidar observation), the layer mean particle extinction coefficient can be determined. These values should be presented as well.

P4, L13: Please explain EOS AM-1

P4, Section 2.4: Isn't that a similar study of OMPS UVAI maps as already presented by Khaykin et al. (2018, in the supplementary part)? Should be mentioned.

P5, L3: I see a clear and strong jump in the aerosol load from 11 to 13 August 2017 over the northern parts of western North America (Canada), in Figure 4. This is a clear and convincing indication that the pyrocumulonimbus cluster activities on 12 August were most probably responsible for these aerosol features. But this not explicitly written.

P5, L10: explain AIRS!

P5, L10-25: The same holds for Figure 5, a clear and sudden increase in the CO concentration from 11 to 13 August is visible. Furthermore, this 12 August event seems to be the reason for the extreme smoke load over Europe on 21-22 August (Figure 5f) as reported by Ansmann et al. (2018). This should be also mentioned.

P5, CALIPSO section 2.6: You already discuss the linear depolarization ratio, but the definition and explanations are provided later (in section 3). This is one of the reasons, why you better start with a more technical section right after the introduction.

P6, L3: Mixed-phase clouds can produce depolarization ratios from almost zero (mainly drops) to 40% (mainly ice crystals), so why should they show depol values of 26-35% only?

P6, L8: Why does an increasing trend in the depolarization ratio indicate aging of smoke particles? First of all, I do not see a trend in the noisy CALIPSO measurements. And second, the strong depolarization can be simply explained by the irregular shape of the smoke particles…, and why should the irregular shape change with time in the stratosphere where dry soot particles seem to dominate.

P6, section 3: Interrupt! A complex section is given, which is only interesting for lidar experts. But the paper is written for the research community dealing the atmospheric smoke and aerosols.

In Sect. 2.1, the lidars are explained (Raman lidars). And in Section3, we learn that you make only use of the Klett method. This makes quite a significant difference. The lidar ratios at 355 and 532 nm are not directly measured, you need to assume that the lidar ratio is height constant (and equal to the obtained layer mean value), so the solutions of the backscatter and extinction coefficients at a given wavelength are not independent of each other, the profiles are rather similar. And at the end, you use the non-independent extinction and backscatter solutions in the lidar inversion retrieval to obtain the microcphysical properties. This inversion algorithm however needs independently measured backscatter and extinction coefficients at 355, 532 and 1064 nm (backscatter). Please mention and explain this 'contradiction' in more detail. What are the consequences for the inversion uncertainties?

P7, L7: You calculate the signal means (within 500 m thick layers) at aerosol layer top and base. Please be precise. You probably use signals below the base and above the top only? What happens if there are traces of smoke above and below the layer, what uncertainty causes the use of noisy Rayleigh signals above and below the aerosol layer in the lidar ratio retrieval? All in all, be using the simple Klett method the uncertainty in the lidar ratio values (at 355 and 532 nm) must be at least 30%. All this needs to be clearly stated.   We need overall uncertainties in the presented lidar ratios at the end of section 3.1.1.

The same for section 3.1.2: We need overall uncertainties in the depolarization values at the end of the section. If you have already 10% uncertainty in the volume depolarization ratio at 355 and 532 nm, the uncertainties in the particle linear depolarization  ratios will be close to 20% for the particle depolarization ratio (especially at 355 nm). How trustworthy are such very large 355 nm particle depolarization ratios of 28% you present later on in the article? Even in the case of desert dust (rather irregularly shaped particles) the particle depolarization ratio is usually 25% or less. Please comment on this.

P9, Sect. 4: First question: The given standard deviations (in the text and in Table 1) indicate the uncertainty or the atmospheric variability (in time and in the vertical profile) or the uncertainty in the retrieval? It is not clear to me. Please state that clearly.

Table 1: I miss the information about the height (base, top) of the smoke layers in the stratosphere in the table. I would recommend to provide the layer-mean extinction coefficients as well.

P10, L15: If you have an uncertainty of 20% in the 355 nm backscatter value and 10% in the volume depolarization ratio then it is hard to get an uncertainty of 15% in the particle depol ratio. I ask this to force you to re-check the results concerning the very large 355 nm particle depolarization ratios of 28%.

P10, L29, Figure 8 and P11, L5: The only reliable profiles are the Klett solutions for the backscatter profiles. The extinction profiles and the EAE profiles are estimates and are strongly based on the used height-constant (layer mean)  lidar ratios. It should be made very clear that the solutions of backscatter and extinction are so similar (or better identical in the profile characteristics) because of the use of the Klett method and the assumption of a height-constant lidar ratio.

In the discussion of the Angstrom values (AE) you may know that EAE = BAE +LRAE, as given in Ansmann et al. (JGR, 2002 on ACE-2 observations in Portugal).  And that fits very well in your observational cases. However, if the assumed LRAE is height constant, and the BAE is height constant, EAE can only be height constant. But in reality, EAE and LRAE may vary. This should be mentioned. Without the Raman lidar approach you cannot provide information on the 'real world' EAE, extinction, and lidar ratio profiles. They are based on an assumption, and not on measured facts. So it is misleading to show profiles of extinction and EAE without saying that these are just retrieval products heavily controlled by the assumption of a height constant lidar ratio.

P10, L30: 'The UTLS aerosol layer was between 17 and 18 km'.  This is one of the sentences that forced me to ask: Why do you call that an UTLS layers. The layer is clearly in the lower stratosphere.

P11, L13: Figure 10 is questionable and probably not representative to explain the long range transport including transport times. The CALIPSO Figure 6 tells us that the smoke layers were mostly at heights of 14-17 km over North America and the Atlantic, sometimes even below 14 km. And at these heights the wind speeds were much higher than at 17-19 km (in Figure 10). We need more trajectories! Especially for the CALIPSO heights (12-17 km) before we can make conclusions on the travel duration (in the discussion section 5). And one effect makes the use of trajectories difficult: The ascent of the layers from day to day by absorption of sunlight! This is not considered in any trajectory modeling. So trajectories are of limited use here. But 20 days of travel from Canada to Europe? That would be quite new, in view all the papers  on Canadian smoke and long range transport. Even Khaykin et al. (2018)

writes that the smoke needed 21 days to travel around the globe (at midlatitudes) to be back over Canada again, and that the smoke needed only 10 days or less to reach Europe.

P11, section 4.2.2: Again the question: How trustworthy are the solutions of the inversion method when only based on Klett optical properties, so that the basic information are backscatter profiles? Please comment on that.

Discussion section:

P12, L26-28: I have my doubts that one singular event (the major pyrocumulonimbus cluster on 12 August) leads to so many smoke layer that you observed between 24-31 August. To my opinion, layers observed 10 days later maybe linked to the 12 August event, and maybe even layers observed on 24 August. But all other smoke layers later on were most probably triggered by other reasons and causes. And note that the absorbing smoke layers ascended during the travel (because of strong absorption of solar radiation). Khaykin et al. (2018) mentioned, 2-3 km per day during the first days. So layers can easily reach the lower stratosphere after 3 days when initially injected to 8 km height ….

P13, L1: In the discussion of the very large 355 nm particle depol values, please check: There is a corrigendum note to the Burton 2015 paper (you will find it on the ACP page). In this corrigendum note it is stated that the 355 nm depolarization ratios were only 20.5% for the smoke case (and not 24% as erroneously obtained in the beginning of the data analysis and published in Burton et al. (2015) paper). So, again, the 28% you get at 355 nm are really 'outstanding' and must be re-checked.… Find out, for example, how large the impact of the Klett backscatter profile is in the retrieval of the particle depol ratio at 355 nm… by using plus/minus 10% backscatter coefficient profiles.

P13, L5-L22: You provide the suggestion (Figure 11) that the depolarization ratio increases with travel time. This is surprising, because Nisantzi et al. (ACP, 2014, Injection of mineral dust into the free troposphere during fire events observed with polarization lidar at Limassol, Cyprus) find the exact opposite and show that in a figure: Decreasing depolarization ratio with transport time. However, their study is exclusively based on tropospheric smoke. And you combine tropospheric and stratospheric observations in your analysis.

Who is right? Maybe both! But first of all, one has to clearly distinguish aerosols in the troposphere and in the stratosphere. As mentioned, tropospheric as well as stratospheric smoke depolarization ratios are considered in your Figure 11. To my opinion, in this way you compare apples and oranges, and therefore the conclusions maybe wrong, at least are 'dangerous'. In the troposphere, coagulation and interaction with gases and moisture can take place, as you discuss so that aging leads to growing particles. According to measurements they get coated (Dahlkoetter et al, ACP 2014), they get coated with liquid stuff, so they get more and more spherical. And that influences the depolarization ratio. The depol ratio decreases, in accordance with the Haarig et al. (2018) results. In the stratosphere all this is less probable, and coating of soot particle is practically suppressed because the moisture and all the gases are not given in the stratosphere. It seems to be that the particles in the stratosphere remain unchanged during the travel. That is what I see also from the noisy CALIPSO observations in Figure 6 and what is also shown in the Haarig paper. But maybe I am wrong. Please provide us with a convincing explanation why aging leads to an increase in the depolarization ratio!

Disregarding my personal opinion, as a consequence of the statements above and the rather different particle aging processes in the troposphere and stratosphere…: If you want to show Figure 11 than you

need to clearly indicate the stratospheric values, may be by very different symbols, e.g., by big open circles. We need a strong contrast between tropospheric and stratospheric values… The lower values in the Figure are obviously tropospheric depol values (apples) , and the higher ones are stratospheric values (oranges) . Furthermore I have my doubts about transport times of more than 14 days…. The related trajectories are always very uncertain and thus questionable.

Final point: I think the primary goal of the discussion section should focus on the comparison with other findings, especially with Haarig et al. (2018), and also Burton et al. (2015, upper tropospheric smoke case). As mentioned several times, your results are based (to a large fraction) on the use of the Klett method. And this includes the determination of the particle depolarization ratios because the Klett backscatter profiles are needed. This can be a significant reason for deviations, especially in the case of the products at 355nm.

P13, L23: Regarding the laboratory studies: Miffre et al (2016) used Arizona Test Dust (ATD). This dust is not a natural dust component. It is 'manufactored' by a company and ATD particles have rather sharp edges. As a result, ATD shows larger depol ratios then usual desert dust particles.

Regarding the Jaervinen et al. (2016) study: These authors made their observations by one wavelength (488 nm, and partly at 552 nm) only! ..and not at 355 and 532 nm, as you mention it. But they showed their observations as a function of the size parameter (size mode diameter /wavelength) so that Mamouri and Ansmann (2016) estimated the probable depolarization ratios for 355, 532, and and 1064 nm. The results (depol ratio for fine dust at 355, 532, and 1064 nm derived from the Jaervinen study) are shown in Haarig et al. (2018). Almost all your discussion information on page 13 (L23-34) was discussed in the same way in the papers of Mamouri and Ansmann (2014, 2016). So, please add these papers to your references.

P14, L37: Regarding the effective radius (from the lidar inversion retrieval), please compare with Haarig et al. (2018). Is there agreement? Please discuss the comparison.

P14, L39: Again, please avoid mixing of tropospheric and stratospheric effects of potential particle growth. Müller et al (2007b) focusses on tropospheric smoke only when discussing aging effects.

P15, conclusion section: The results of the paper are summarized only in this section. But concluding remarks are not given. So maybe, create some concluding remarks, some outlook ideas…

---

## Referee Comment (RC1) · Anonymous Referee #1 · 4 Aug 2018

GENERAL COMMENT

The paper presents an interesting study on long-range transported smoke aerosols, originated in Canada wildfires, in the UTLS (Upper Troposphere/Lower Stratosphere) over Europe detected at several EARLINET stations in summer 2017 in combination with satellite observations. Optical depth at 532 nm from 0.05 to above 0.20 were detected, with very weak spectral dependence. Other particle microphysical properties like Lidar ratio and particle depolarization ratios suggest the presence of aged smoke likely with complicated morphology. The retrieved aerosol properties allowed the computation of the direct radiative forcing (DRF) effect originated in the UTLS aerosol lay-

ers and the radiative heating rates in this layers that are coherent with the observed radiosonde temperature profiles. The paper is worthy to be published in ACP having in mind that evidences the capabilities of a lidar network focused on tropospheric research in obtaining valuable information on the UTLS aerosols. For this purpose both the advanced instrumentation and the analytical tools used are crucial. The paper is well written and offers valuable information for the reader. Nevertheless the clarification of some points will enhance the quality of the paper.

PARTICULAR COMMENTS

In this work it is especially relevant to get information on the accuracy and uncertainties of the retrievals. The AOD retrievals and the lidar ratio retrievals are clearly related in the analysis procedure used. In this sense, the approach followed for the computation of the UTLS AOD and Lidar ratio with the fixed lidar systems is stated and details on the error propagation and discussion on the accuracy and uncertainty of the retrievals is presented. Nevertheless, some points require additional clarification. Thus, concerning the discussion on the error propagation, in Page 7, I have a question: Are the authors assuming the absence of errors in the molecular part? Having in mind the impact of an accurate thermodynamic profile on this assumption I do not see any information on the thermal profile used. Furthermore, although the computation of the uncertainties is applied in the analysis sections, it would be worthy to include some quantitative information concerning the final uncertainties of the UTLS AOD and UTLS lidar ratio retrievals in the last paragraph in section 3.1.1.

In the case of the MAMS lidar retrievals there are additional limitations and the issue of accuracy and uncertainty is particularly relevant. Thus, in spite of the auxiliary use of its data, it is necessary to include additional discussion on the reduced accuracy of this retrievals.

Considering the uncertainty in the AOD retrievals the AOD spectral dependence will present a large uncertainty that requires additional discussion.

More details about the procedure used in GARRLIC for the computation of the aerosol DRF and the UTLS layer heating rates must be provided.

Minor changes

The references on the EARLINET network must include a recent reference that updates the features of the network: Pappalardo, G., Amodeo, A., Apituley, A., Comeron, A., Freudenthaler, V., Linné, H., Ansmann, A., Bösenberg, J., D'Amico, G., Mattis, I., Mona, L., Wandinger, U., Amiridis, V., Alados-Arboledas, L., Nicolae, D., and Wiegner, M.: EARLINET: towards an advanced sustainable European aerosol lidar network, Atmos. Meas. Tech., 7, 2389-2409, https://doi.org/10.5194/amt-7-2389-2014, 2014.

Please consider the following reformulation of the statement on Page 6 Line 2 "at this temperature clouds consist mainly of ice crystals"

In order to increase the clarity of the test include the following changes in the first paragraph of section 3.1.1: Substitute:" The integral of the extinction coefficient over the UTLS layer, expressed below, is compared with the pre-calculated optical depth" by "The UTLS AOD is calculated by the integral of the extinction coefficient over the UTLS layer, expressed below". And after equation (2) reformulate the statements: "This pre-calculated optical depth is derived from the elastic channel at 355 and 532 nm. The method is widely used in cirrus clouds studies (Platt, 1973; Young, 1995). " as follows" by "This derived value of AOD is compared with the pre-calculated optical depth obtained from the elastic channel at 355 and 532 nm using a method widely used in cirrus clouds studies (Platt, 1973; Young, 1995)"

In page 7 Line 7 change: "We calculate the signal mean within a window of 0.5 km..." by "We calculate the lidar signal mean within a window of 0.5 km. . ."

In page 9 line 13 consider changing "intervals" by "periods".
* * *

---

## Referee Comment (RC2) · Anonymous Referee #2 · 8 Aug 2018

General assessment and major comments

This study provides lidar measurements from two sites in Northern France, showing long range transported smoke in the UTLS. The absorbing nature of smoke is crucial for the stratospheric height ranges, concerning both heating rates (HR) and direct radiative forcing (DRF). The authors try to estimate the DRF and HR and their results show decrease of the radiation reaching the surface and an increased HR due to the absorption of the solar radiation at TOA. In general, I find this study very interesting and of high value. It is a study that fits well in the EARLINET special issue, since it demonstrates the value of EARLINET lidars for atmospheric research in both troposphere and

stratosphere. However, before proceeding with publication in ACP, I strongly suggest that the authors would revise the following points:

1. Page 11, Lines 18-27: "The spheroid model was used to retrieved dust properties (Dubovik et al., 2006; Mishchenko et al., 1997; Veselovskii et al., 2010). But it is not clear if this model is applicable to soot particles with complicated morphology. The size of smoke particles is expected not too big so that we choose to apply regluarization algorithm with sphere model." The retrieved microphysical properties seem to be associated with high uncertainties, since the shape used (spherical) does not reproduce the depolarization measurements and it should not reproduce accurately the backscattered light measurements either. The reported uncertainties in Table 2 refer to cases of spherical particles and are not representative. Please provide a better assessment of the retrieval uncertainties.

2. Regarding the DRF calculations: these are based on the retrieved microphysical (point 1) properties which, as discussed above, are derived from the 3b+2a regularization inversion and are associated with (most probably) high uncertainties. Especially for the imaginary part this uncertainty is expected to be the highest (Burton et al., 2016). Please provide a better assessment of the retrieved property uncertainties and quantify the uncertainties of the DRF calculations accordingly. If this is not possible, omit section 4.2.3 from the manuscript. This also applies to Page 14, Lines 9-13, where the derived complex refractive index is compared to other studies. Omitting 4.2.3 would not affect the quality of the paper, since the authors already provide important results on smoke optical properties and microphysical estimates.

3. Another issue addressed in this study is the increase in particle depolarization ratio at 532 nm which is attributed to the particle aging. The authors gathered observations of the particle linear depolarization at 532nm from previous studies and have also included the results obtained from the present study. Nevertheless, the only visible trend seen in Figure 11 results from CALIPSO measurements. From the ground-based lidars in Lille and Palaiseu, there is no obvious increase at 532nm. In conclusion, the

phrase"we found an increase in depolarization versus transport time" in the manuscript abstract should be changed to "CALIPSO observations of the UTLS smoke layer suggest an increase in depolarization at 532nm versus transport time".

Minor comments

Page1, Line 9: "Typical particle depolarization" the meaning of the word typical should be clarified by the authors, meaning what is the definition of linear particle depolarization ratio used? (is it the cross/parallel ratio or the cross/total ratio?)

Page 1, Line 10: "The relatively high depolarization ratios and such spectral dependence are an indication of a complicated morphology of aged smoke particles" The conclusion that the spectral dependence of the depolarization ratio is characteristic of aged smoke particles can be hardly drawn by two cases, i.e. the current one and the one reported in Burton et al. (2015). Please rephrase accordingly. Page2, Line 30: "We focus on the retrieval of the aerosol optical and microphysical properties from the Lidar measurements". The authors should highlight that the depolarization ratio values are not reproduced in the retrieval of the microphysical properties. Page 4, Line 2: Please change the phrase "showed an increase of temperature in the stratospheric smoke layers" to "An increase of temperature due to the presence of smoke aerosols in this region" or something similar. Page 5, Line 3: Change the phrase "A plume with relatively high UVAI first occurred over the British Columbia on 11 August, and the intensity of the plume was moderate" to "a plume of moderate intensity and relatively high UVAI, first occurred over British Columbia on 11 August. Page 5, Line 4: Please change the phrase "and the UVAI in the center of the plume reached above 10" to "and the UVAI in the center of the plume reached above 10, as indicated by the grey area on the plot (Fig 4)" Page 6, Line 1: "We have examined the temperature profiles" Did you use radiosonde measurements? Please provide more info. Page 6, Line 2: "the temperature drops below -38_C, at which temperature the cloud droplets mostly turn to ice phase" Please provide relevant reference.

Page 6, Line 8: "The increasing trend of the depolarization ratio is probably due to aerosol aging" As discussed above, this is a hardly drawn conclusion. Please rephrase accordingly.

Page 7, Line 7: "we can calculate the optical depth of the cirrus cloud" Please change "cirrus cloud" to "UTLS aerosol layer" since this is what you refer to in this case.

Page 7, Line 12: change the phrase "are considered as the major error sources of the optical depth" to "are considered as the major error sources in the estimation of the optical depth"

Page 7, Line 13: "based on the statistical error of photon distributions" Please provide more info on the definition of the noise of your lidar measurements. Do you take into account the systematic errors?

Page 7, Line 22: change the phrase "of the error of optical depth" to "of the error of optical depth to the estimation of the Lidar ratio"

Page 10, Line 15: typing error, change volume depolarization ratio at 355 nm to molecular depolarization ratio at 355 nm.

Page 14, Line 14: "Smoke in dry conditions have higher refractive indices than that in wet condition" Provide relevant reference.

Figure 6: The x axes on CALIPSO plots should be the same in order to show the variation. Also the phrase in the caption "The profiles of backscatter coefficient and particle linear depolarization ratio (PLDR)" could be changed to "The profiles of backscatter coefficient and particle linear depolarization ratio (PLDR) at 532nm from CALIPSO" Figure 7: The points on this figure should be larger to be more visible. Also, it would be better if the colors of the points are different for the two lidar systems.

---

## Referee Comment (RC3) · Anonymous Referee #3 · 17 Aug 2018

The paper provides unique measurements of an extreme event of smoke advection from Canada to Europe. Smoke could be observed within the Troposphere as well as in the Stratosphere. I consider the measurements and the resulting data as very valuable and of interest for the scientific community and the topic fits well in the scope of ACP.

However, in my opinion the paper tries to cover too much topics at once (optical properties of smoke, microphysical properties, source analysis, change of depolarization with aging time by using Calipso data, radiative transfer calculations, estimating of heating rates, temperature changes in the stratosphere etc.). Thus, the paper is partly confus-

ing and for some of the topics the proper fundament needed for the conclusion drawn are missing.

I therefore recommend first to focus on your key expertise and present only the unique measurements, which are already shown and exploit as much as possible (optical and inversion results). Of course you should also include the satellite data for source analysis. However please focus on the lidar measurments in France.

Then, in another paper(s), the other topics could be covered (i.e. radiative transfer calculations and heating rates etc.) Especially the discussion of the impact of the smoke plumes need much more detail and evidence, as for example it is not shown nor discussed that temperature increases in the stratosphere at a certain height are not caused by the complicated upper atmospheric circulation including tropopause foldings etc. I therefore recommend for the second paper to include some expertise of the upper atmospheric dynamics and probably some modelling to show and prove the impact of the smoke layer with its possible warming on the overall general condition of the UTLS region above Europe.

I also think that the use of Garrlic/Grasp radiative transfer code to obtain radiative impact must be explained much better. Currently, it is not understandable how the calculations are performed.

From the current version, I also doubt for the evidence of the increase in smoke depolarization with aging time. I think the use of Calipso data for such a study is again a topic of its own. Uncertainties in the Calipso retrieval (see comments in pdf) should be discussed. Even more important, other influencing factors as the relative humidity, the altitude of the smoke with respect to ground level but also tropopause should be investigated. Among others influencing factors (fire source, burning types).

I therefore recommend major revision with the recommendation written above.

Find other comments below and as pdf-comments in the supplement.

Paper structure:

-The abstract is too long. Please shorten.

-The introduction is in my opinion a loose sequence of different paragraphs and therefore not constructive. Please revise and make it more focusing on your topic. E.g.: have such events been reported earlier? Is this the first time? . . .

-Several facts concluded in the observations section are again raised in the discussion. I think you should shape your paper. Either describe your observations only, and then make a discussion in a separate section or make conclusions also in the observations section but then discuss only new issues in the discussion section.

Terminology

-I would recommend not to use "upper troposphere/lower stratosphere UTLS" as a standard the term. This historical term covers all altitudes between 5 and 30 km and thus does not make clear that a significant portion of the smoke you detected is above the local tropopause. Please feel free and state, that there is stratospheric smoke as well as smoke in the free troposphere. In most times you anyhow refer to stratospheric smoke with your statements. . .

Major scientific remarks:

-What about the uncertainties of Calipso. For example for the PLDR, the particle backscatter coeff. is needed which is in turn calculated with a Klett-like approach, i.e. a a-priori lidar ratio. Thus the questions arises, which aerosol type was classified by Caliop and which lidar ratio was used to obtain the PLDR and is this correct for your aerosol type of investigation.

-Depolarization: The molecular depolarization ratio depends on your filters used in the lidar. Are the theoretical values you stated valid for your system? And can you neglect temperature effects? Which molecular depol ratio value did you use for the PLDR calculation? The measured one or the theoretical one?

-Radiative transfer: The methodology explanation is too short. It is not reproducible how you performed the radiative transfer. Which parameters did you use as input? Which are constrained by the algorithm? and so on... thus, either you introduce a much more detailed description of this method and the paper gets longer or you shift these calculations to a second paper.

-Inversion: Is it useful to perform an inversion when having only 3 elastic signals? I mean lidar ratio is not an independent variable in your case...please discuss this! What justifies using a sphere model when particle size is small.

-Increase in stratospheric temperature. Your explanations are not convincing concerning the temperature increase. Did you consider also all other processes in the UTLS? May this only be normal variability? How long do you observed a temperature increase? So please add more detail or shift to another paper (I think this is a topic alone).

-Depolarization ratio vs. aging time: The provided graphic and literature review does not convince me of the given causality, please also investigate the RH, the height etc. vs depolarization ratio.

Please also note the supplement to this comment:
https://www.atmos-chem-phys-discuss.net/acp-2018-655/acp-2018-655-RC3-supplement.pdf

**Supplement:**

[Figure]

**A study of long-range transported smoke aerosols in the Upper Troposphere/Lower Stratosphere**

Qiaoyun Hu[1], Philippe Goloub[1], Igor Veselovskii[2], Juan-Antonio Bravo Aranda[3], Ioana Popovici[1,4], Thierry Podvin[1], Martial Haeffelin[3], Anton Lopatin[5], Christophe Pietras[3], Xin Huang[5], Benjamin Torres[1], and Cheng Chen[1]

[1]Univ. Lille, CNRS, UMR8518 – LOA – Laboratoire d'Optique Atmosphérique, 59000 Lille, France
[2]Physics Instrumentation Center of GPI, Troitsk, Moscow, 142190, Russia
[3]Institut Pierre Simon Laplace, École Polytechnique, CNRS, Université Paris-Saclay, 91128 Palaiseau, France
[4]CIMEL Electronique, 75011 Paris, France
[5]GRASP-SAS, Remote sensing developments, 59650 Villeneuve d'Ascq, France

**Correspondence:** Qiaoyun Hu (qiaoyun.hu@univ-lille.fr)

**Abstract.** Long-range transported smoke aerosols in the UTLS (Upper Troposphere/Lower Stratosphere) over Europe were detected in Summer 2017. The measurements of ground-based instruments and satellite sensors indicate that the UTLS aerosol layers were originated from Canadian wildfires and were transported to Europe by
[Figure]
TLS advection. In this study, the observations of two multi-wavelength Raman Lidar systems in northern France (Lille and Palaiseau) are used to derive aerosol properties, such as optical depth of the UTLS layer, Lidar ratios at 355 and 532 nm and particle linear depolarization ratios at 355, 532 and 1064 nm. The optical depth of the UTLS layers at 532 nm varies from 0.05 to above 0.20, with very weak spectral dependence between 355 and 532 nm. Lidar ratios at 355 nm are in $31 \pm 15$ sr to $45 \pm 9$ sr range and at 532 nm, the Lidar ratios are in the range of $54 \pm 12$ sr to $58 \pm 9$ sr. Such spectral dependence of Lidar ratio is known to be a characteristic feature of aged smoke. The typical particle depolarization ratios in the UTLS smoke layer are $25 \pm 4\%$ at 355 nm, $19 \pm 3\%$ at 532 nm and $4.5 \pm 0.8\%$ at 1064 nm. The relatively high depolarization ratios and such spectral dependence are an indication of a complicated morphology of aged smoke particles. We found an increase of depolarization ratio versus transport time. The depolarization ratio at 532 nm increases from below $2 - 5\%$ for fresh smoke to over 20% for smoke aged more than 20 days. The $3\beta + 2\alpha$ observations of two cases at Palaiseau and Lille sites were inverted to the aerosol microphysical properties using *regularization* algorithm. The particles distribute in the 0.1–1.0 $\mu$m range with effective radius of $0.33 \pm 0.10$ $\mu$m for both cases. The derived complex refractive indices are $1.52(\pm0.05) + i0.021(\pm0.010)$ and $1.55(\pm0.05) + i0.028(\pm0.014)$ for Palaiseau and Lille data. The retrieved aerosol properties were used to calculate the direct radiative forcing (DRF) effect specific to the UTLS aerosol layers. The simulations derive daily net DRF efficiency of -79.6 Wm$^{-2}\tau^{-1}$ at the bottom of the atmosphere for Lille observations. At the top of the atmosphere, the net DRF efficiency is -7.9 Wm$^{-2}\tau^{-1}$. The results indicate that the UTLS aerosols strongly reduce the radiation reaching the terrestrial surface by absorption. The heating rate of the UTLS layers is estimated to be 3.7 $Kday^{-1}$. The inversion of Palaiseau data leads to similar results. The heating rate predicts a temperature increase within the UTLS aerosol layer, which has been observed by the radiosonde temperature measurements.

**Number: 1**  Author:  Subject: Comment on Text  Date: 17.08.2018 14:01:33
? what is this ? Is this a specific scientific term?

**Number: 2**  Author:  Subject: Comment on Text  Date: 17.08.2018 14:01:55
all in stratosphere?

**Number: 3**  Author:  Subject: Comment on Text  Date: 17.08.2018 10:17:58
Can you prove that these findings are caused by the smoke?

[Figure]

**1 Introduction**

UTLS lies between the middle troposphere and the middle stratosphere. UTLS aerosols play an important role in the global radiative budget and chemistry-climate coupling (Deshler, 2008; Kremser et al., 2016; Shepherd, 2007). According to the in-situ measurements provided by PALMS (Particle Analysis by Laser Mass Spectrometry) instrument (Murphy et al., 2007, 2014),

5  UTLS aerosols are categorized into three branches: sulphuric acid with metals from ablation of meteoroids, nearly sulphuric acid with associated water and organic-sulphate particles, as well as a small fraction of dust, sea salt and other types originating from the troposphere. The volcanic eruption is considered as the most significant contribution of stratospheric aerosols because the explosive force could be sufficient enough to penetrate the tropopause, which is regarded as a barrier to the convection between the troposphere and stratosphere.

10  Lidar is an important tool in studying and profiling the UTLS aerosols. Hofmann et al. (2009) presented an increase of background stratospheric aerosols observed by Lidar system. The study showed the aerosol backscattering in the altitude range 20–30 km increased by 4–7% per year. Khaykin et al. (2017) studied the variability and evolution of the midlatitude stratospheric aerosols using 22 years of ground-based Lidar and satellite observations. This study provided an indication of a growth in the non-volcanic component of stratospheric aerosol over the last two decades. Zuev et al. (2017) presented 30-year Lidar

15  observations of the stratospheric aerosol layers coming from volcanic eruptions, polar stratospheric clouds and strong wildfire emissions over western Siberia.

In the summer of 2017, intense wildfires have spread in the west and north of Canada. By mid-August, the burnt area had grown to almost 9000 km$^2$ in British Columbia, which broke the record set in 1958 (see the link). The severe wildfires generated very strong pyro-cumulonimbus clouds (see the link) which were recorded by the satellite imaginary MODIS (Moderate Resolu-

20  tion Imaging Spectrometer). The pyro-cumulonimbus clouds are an extreme form of the pyro-cumulus clouds which have the potential to transport the fire emissions from the planetary boundary layer to the UTLS (Luderer et al., 2006; Trentmann et al., 2006), thus affecting the atmospheric chemistry of the stratosphere. Previous studies have (Fromm et al., 2000; Fromm and Servranckx, 2003) reported smoke aerosols lofted to the UTLS. The GOES-15 (Geostationary Operational Environmental Satellite) detected five pyro-cumulonimbus clouds over the British Columbia on 12 August 2017 (see the link).

25  Reoccurring aerosol layers in the UTLS were detected by the Lidar systems in northern France during 19 August and 12 September 2017. In this study, we present the Lidar observations from two French Lidar stations: Lille (50.612°N, 3.142°E, 60 m a.s.l) and Palaiseau (48.712°N, 2.215°E, 156 m a.s.l). Satellite measurements from multiple sensors, including UVAI (Ultraviolet aerosol index) from the OMPS NM (Ozone Mapping and Profiler Suite, Nadir Mapper), CO concentration from AIRS (Atmospheric Infrared Sounder), backscatter coefficient and depolarization profiles from CALIPSO (Cloud-Aerosol Lidar and

30  Infrared Pathfinder Satellite Observations) help identify the source and the transport pathway of the UTLS layers. We focus on the retrieval of the aerosol optical and microphysical properties from the Lidar measurements. Further, we study the radiative effect of the UTLS aerosol layer.
* * *
**Number: 1** Author: Subject: Comment on Text Date: 17.08.2018 14:02:18

where have these measurements been made? during aircraft flights? at which locations?
* * *
**Number: 2** Author: Subject: Comment on Text Date: 17.08.2018 14:06:58

I think this statement is in contradiction to your definition of the UTLS. As you know, in the troposphere above 5 km a lot of different aerosols can be found (see e.g. all the EARLINET activities). I think this statement is mainly valid for the stratosphere. I therefore recommend not to stick to the UTLS term, but refer to the stratosphere. You did this already in the next sentence, so staying with this term is really confusing....
* * *
**Number: 3** Author: Subject: Comment on Text Date: 17.08.2018 10:26:11

What is the motivation of presenting this information in context to your study?
* * *
**Number: 4** Author: Subject: Comment on Text Date: 17.08.2018 10:27:54

not working
* * *
**Number: 5** Author: Subject: Comment on Text Date: 06.08.2018 12:58:33

links not working
* * *
**Number: 6** Author: Subject: Comment on Text Date: 17.08.2018 10:28:16

? The sentence is basically the same as the one before but with different citations.
* * *
**Number: 7** Author: Subject: Comment on Text Date: 17.08.2018 14:07:23

I think in this paragraph you should also refer to the publications which covers already this specific topic.

[Figure]

**2   Ground-based and satellite observations**

**2.1   Simultaneous Lidar and sun photometer observations**

[revised manuscript text omitted]

**Number: 1** Author: Subject: Inserted Text Date: 17.08.2018 10:39:19

obvious

**Number: 2** Author: Subject: Comment on Text Date: 17.08.2018 10:40:22

tropopause folding considered? how is the normal climatology at this height? What about the normal stratospheric dynamics and variability. Please discuss best in second paper.

**Number: 3** Author: Subject: Cross-Out Date: 17.08.2018 10:41:07

**Number: 4** Author: Subject: Cross-Out Date: 17.08.2018 10:40:23

**Number: 5** Author: Subject: Comment on Text Date: 17.08.2018 10:42:07

reference or is this your statement?

**Number: 6** Author: Subject: Comment on Text Date: 17.08.2018 10:43:52

why 15 August, please explain.

**Number: 7** Author: Subject: Comment on Text Date: 17.08.2018 10:42:58

? what does it mean? Are you want to state that this is visible smoke?

**Number: 8** Author: Subject: Comment on Text Date: 17.08.2018 10:44:42

? ever explained?

[revised manuscript text omitted]

**Page: 6**
* * *
**Number: 1**     Author:     Subject: Comment on Text     Date: 17.08.2018 14:34:17

these values could also yield for smoke, dust etc....thus probably not a good indicator for aerosol....these values are anyhow not needed, as you state that temperatures were below -38 °C.
* * *
**Number: 2**     Author:     Subject: Comment on Text     Date: 17.08.2018 14:35:38

only when assuming that the Klett retrieval was performed with the correct lidar ratio, i.e. for clouds or smoke....
However as I suspect that it is not the case, please clarify how this layer was classified by calipso, and/or plot volume depolarization ratios to continue your argumentation
* * *
**Number: 3**     Author:     Subject: Comment on Text     Date: 17.08.2018 14:14:30

please check literature for backscatter values of mixed phase clouds, usually these are much higher. What is the outcome of the caliop scene classification? cloud or not? which particle type was identified?
* * *
**Number: 4**     Author:     Subject: Comment on Text     Date: 17.08.2018 14:35:59

please define what is UTLS advection ....
* * *
**Number: 5**     Author:     Subject: Inserted Text     Date: 17.08.2018 11:08:42

as explained in the following.
* * *
**Number: 6**     Author:     Subject: Inserted Text     Date: 17.08.2018 11:09:13

(methodology described below)
* * *
**Number: 7**     Author:     Subject: Cross-Out    Date: 06.08.2018 14:04:39
* * *
**Number: 8**     Author:     Subject: Comment on Text     Date: 17.08.2018 11:13:29

how valid is your assumption, please discuss
* * *
**Number: 9**     Author:     Date: 06.08.2018 14:04:52

aerosol layer

cirrus clouds,
 below:

$$\tau^u(\lambda) = \frac{1}{2} \, ln \frac{\overline{P}_{base}(\lambda) \, r_{base}^2 \, \beta_m(\lambda, r_{top})}{\overline{P}_{top}(\lambda) \, r_{top}^2 \, \beta_m(\lambda, r_{base})} - \int\limits_{r_{base}}^{r_{top}} \alpha_m(\lambda, r) dr \tag{3}$$

where $\tau^u$ is the optical depth of the UTLS aerosol layers. $\overline{P}_{top}$ and $\overline{P}_{base}$ represent the mean Lidar signal at the top and the base of the UTLS layer. $\alpha_m$ and $\beta_m$ are the molecular extinction an backscatter coefficient. We use this method to estimate the optical depth of the UTLS layer for LILAS and IPRAL measurements. The Lidar ratio leading to the best $\tau^i$ and $\tau^u$

5  is accepted as the retrieved Lidar ratio of the UTLS aerosol layer. We apply Klett inversion only to the UTLS aerosol layer, therefore, the impact of tropospheric aerosols is excluded.

We calculate the signal mean within a window of 0.5 km at the top and the base of the aerosol layer, to get $\overline{P}(r_{top}, \lambda)$ and $\overline{P}(r_{base}, \lambda)$. However, this method is not applicable to the MAMS Lidar measurements due to the insufficient signal-to-noise ratio above the UTLS plume. or the MAMS Lidar measurements, the columnar AOD measured by PLASMA sun photometer

10  is used as a constraint. Klett inversion is performed to the Lidar profile from the surface to the top of the UTLS layer, assuming a vertically independent Lidar ratio.

The errors in the Lidar signal at the top and the base of the UTLS layers are considered as the major error sources of the optical depth. We estimate the error of the Lidar signal $\overline{P}(\lambda, r_{top})$ and $\overline{P}(\lambda, r_{base})$ to be 3–5%, based on the statistical error of photon distributions. According to Equation 3, the error of the optical depth, $\frac{\Delta \tau^u}{\tau^u}$, is written as:

$$\left(\frac{\Delta \tau^u}{\tau^u}\right)^2 = F_{\overline{P}_{top}} \left(\frac{\Delta \overline{P}(\lambda, r_{top})}{\overline{P}(\lambda, r_{top})}\right)^2 + F_{\overline{P}_{base}} \left(\frac{\Delta \overline{P}(\lambda, r_{base})}{\overline{P}(\lambda, r_{base})}\right)^2 \tag{4}$$

$$F_{\overline{P}_{top,base}} = \left(\frac{\overline{P}(\lambda, r_{top,base})}{\tau^u} \frac{\partial \tau^u}{\partial \overline{P}(\lambda, r_{top,base})}\right)^2 \tag{5}$$

where $\Delta$ represents the absolute error of the quantity behind it.

The error of optical depth propagates into the error of Lidar ratio and vertically integrated backscatter coefficient. Additionally, the error of the Lidar ratio also relies on the step length of Lidar ratio between two consecutive iterations and the fitting error of

20  the optical depth of the UTLS aerosol layer, which can be limited by narrowing the step of the iteration. In our calculation, we use a step of 0.5 sr and achieve the fitting error of optical depth less than 1% which is negligible compared to the contribution of the error of optical depth. However, we can basically estimate the error of the integral of the backscatter coefficient within the UTLS aerosol layer, not the error of the backscatter coefficient profile.

**3.1.2 Particle linear depolarization ratio**

25  The particle linear depolarization ratio, $\delta_p$, is written as:

$$\delta_p = \frac{R \delta_v (\delta_m + 1) - \delta_m (\delta_v + 1)}{R(\delta_m + 1) - (\delta_v + 1)} \tag{6}$$

**Number: 1** Author: Date: 06.08.2018 14:05:21
aerosol layer

**Number: 2** Author: Subject: Cross-Out Date: 17.08.2018 11:13:31

**Number: 3** Author: Subject: Inserted Text Date: 17.08.2018 11:15:13
agreement

**Number: 4** Author: Date: 17.08.2018 11:15:13
lidar ratio of PBL?

**Number: 5** Author: Subject: Comment on Text Date: 17.08.2018 14:37:17
why not simply use the lidar ratio you retrieved with the other method or the ones reported by Haarig et al.? I consider using the total AOD as a constrained much more critical as you have to use one lidar ratio for all heights....you could also use the lidar ratio of the PBL obtained with LILAS or IPRAl for the MAMS Klett retrieval in the PBL.

**Number: 6** Author: Date: 17.08.2018 14:37:36
what about assumption of rayleigh scattering only for the uncertainty estimation?

[Figure]

where $R$ is the backscatter ratio, $\delta_v$ is the volume linear depolarization ratio and $\delta_m$ is the molecular depolarization ratio. $R$ is defined as the ratio of the total backscatter coefficient to the molecular backscatter coefficient. $\delta_v$ is the ratio of the perpendicularly scattered signal to the parallel scattered signal, taking into account a calibration coefficient. The depolarization calibration is designated to calibrate the electro-optical ratio between the perpendicular and parallel channel and is performed following

5    the procedure proposed by Freudenthaler et al. (2009).

According to Equation 6, the error of particle depolarization ratio lies in three terms: the backscatter ratio $R$, volume depolarization $\delta_v$ and molecular depolarization $\delta_m$.
[Figure]

$$\Big(\frac{\Delta\delta_p}{\delta_p}\Big)^2 = F_R\Big(\frac{\Delta R}{R}\Big)^2 + F_{\delta_v}\Big(\frac{\Delta\delta_v}{\delta_v}\Big)^2 + F_{\delta_m}\Big(\frac{\Delta\delta_m}{\delta_m}\Big)^2 \tag{7}$$

$$F_X = \Big(\frac{X}{\delta_p}\frac{\partial\delta_p}{\partial X}\Big)^2, X = R,\ \delta_v,\ \delta_m \tag{8}$$

As the backscatter ratio and the volume depolarization increase, the dependence of particle depolarization ratio on the backscat-

10    ter ratio gets much weaker. In our study, either the depolarization ratio (at 355 nm) or the backscatter ratio (at 1064 nm), or even both, are high enough, so it allows us to conservatively assume a preliminary error level for the backscatter ratio $R$. We simply assume 20% error in the backscatter ratio. The potential error sources of the volume depolarization come from the optics and the polarization calibration. The optics have been carefully optmized and adjusted to minimize the depolarization contamination. After long-term Lidar operation and monitoring of the depolarization calibration, we conservatively expect 10% relative

15    errors in the volume depolarization ratio. The theoretical molecular depolarization ratio is calculated to be 0.4% with negligible wavelength dependence (Behrendt et al., 2002). In the historical record since 2013, LILAS measured molecular depolarization ratios approximately 0.8–1.3% at 532 nm channel, 1.2–1.8% at 355 nm channel and 0.7–1.0% at 1064 nm channel. IPRAL measured molecular depolarization ratio about 2.0% at 355 nm in this study. Molecular depolarization ratios measured by both LILAS and IPRAL system exceed the theoretical value. Regardless of the error in the polarization calibration, the error of

20    molecular depolarization ratio rises mainly from the optics, precisely, the cross-talks between the two polarization channels. The imperfections of the optics cannot be avoided, but a careful characterization is helpful to eliminate the cross-talks as much as possible (Freudenthaler, 2016). In our study, we simply assume 200% and 300% for the error of molecular depolarization ratio measured by LILAS and IPRAL system, respectively. In the following section, the total error of particle depolarization ratio is calculated according to Equation 7.

25    **3.2   Aerosol inversion and radiative forcing estimation**

The $3\beta + 2\alpha$ from Lidar observations can be inverted to obtain particle microphysical parameters. The regularization algorithm is used to retrieve size distribution, wavelength-independent complex refractive indices, particle number, surface and volume concentrations (Müller et al., 1999; Veselovskii et al., 2002). DRF estimation is performed with the retrieved aerosol microphysical properties, the vertical profile of the UTLS plume and surface BRDF (Bidirectional Reflectance Distribution Function)

30    parameters from AERONET. In this study, we apply the forward model of GRASP (Generalized Retrieval of Aerosol and Surface Properties) to calculate the DRF effect of the UTLS aerosol layer. GRASP is the first unified algorithm developed for

Number: 1    Author:    Date: 06.08.2018 14:12:05

ratio

Number: 2    Author:    Subject: Comment on Text    Date: 17.08.2018 11:23:20

really? or do you mean molecular depolarization ratio?

Number: 3    Author:    Subject: Comment on Text    Date: 17.08.2018 11:24:11

I do not understand the meaning of this sentence.

Number: 4    Author:    Subject: Comment on Text    Date: 17.08.2018 11:25:03

what is this? what do you mean?

Number: 5    Author:    Subject: Comment on Text    Date: 17.08.2018 11:27:49

This depends on your filters. So are these values valid for your system? And can you neglect temperature effects?

Number: 6    Author:    Subject: Comment on Text    Date: 17.08.2018 14:16:40

which value did you use for a molecular depol ratio for PLDR calculation? The measured one or the theoretical one?

Number: 7    Author:    Subject: Comment on Text    Date: 17.08.2018 11:29:15

DRF not yet introduced?

[revised manuscript text omitted]

Number: 1        Author:        Date: 06.08.2018 14:30:12
to

Number: 2        Author:        Subject: Comment on Text        Date: 17.08.2018 14:18:14
this is amazing and you should conclude that this indicates the large-scale characteristic of this event. Please also compare the lidar ratios to other publications.

Number: 3        Author:        Date: 06.08.2018 14:30:24
particle depolarization ratios of

Number: 4        Author:        Subject: Comment on Text        Date: 17.08.2018 14:38:39
in my opinion the error discussion should be shifted to the lidar section were it is already partly discussed. After presenting your findings for the smoke layer I think another new error discussion is tiring.

Number: 5        Author:        Subject: Comment on Text        Date: 17.08.2018 14:19:56
???I guess you mean molecular depolarization ratio...

Number: 6        Author:        Subject: Comment on Text        Date: 06.08.2018 14:33:32
in my opinion it is misleading to write the error for the particle depolarization ratio in %. Do you mean it as absolute or relative value?

Number: 7        Author:        Subject: Comment on Text        Date: 06.08.2018 14:36:51
did you assume these values or is this the result of your described procedure?

Number: 8        Author:        Subject: Comment on Text        Date: 06.08.2018 14:38:26
does these values fit to the observations described by Haarig et al?

[Figure]

about $1.0 \pm 0.3$. The particle depolarization ratios decrease as wavelength increases. The vertical variations of the extinction and backscatter-related Ångström exponent and particle depolarization is weak, indicating that the observed aerosol layer is homogenous.

[Figure]

**28 August 2017, Palaiseau**

5  Figure 9 shows the retrieved optical parameters from IPRAL observations at $1920 - 2120$ UTC, 28 August 2017 in Palaiseau. The thickness of the UTLS layer is about 2.5 km, spreading from 17 km to 19.5 km. Klett inversion was applied with estimated Lidar ratio of 36 sr at 355 nm and 58 sr at 532 nm. At 1064 nm the Lidar ratio was assumed to be 60 sr. The maximum extinction coefficient in the layer reached $0.12$ km$^{-1}$ at 532 nm. The extinction-related Ångström exponent between 355 nm and 532 nm is about $-0.06 \pm 0.3$. The corresponding backscatter Ångström exponent is about $1.2 \pm 0.3$. The particle linear
10  depolarization ratio at 355 nm is about $27 \pm 4\%$. The particle linear depolarization ratio at 355 nm, extinction and backscatter-related Ångström exponent between 355 nm and 532 nm do not show evident vertical variations.

15   CALIPSO profile on 19 August, i.e. Figure 6(d) shows an elevated aerosol layer at $16 - 18$ km near this region.

**4.2.2  Microphysical properties**

Regularization algorithm is applied to the vertically averaged extinction coefficients (at 355 and 532 nm) and backscatter coefficients (at 355, 532 and 1064 nm) in Figure 8 and Figure 9. Treating non-spherical particles is a challenging task. Many
20  studies have been done to model the light scattering of non-spherical particles. The spheroid model was used to retrieved dust properties (Dubovik et al., 2006; Mishchenko et al., 1997; Veselovskii et al., 2010). But it is not clear if this model is applicable to soot particles with complicated morphology. The size of smoke particles is expected not too big so that we choose to apply regluarization algorithm with sphere model. The particle linear depolarization ratio is not used in the retrieval, and the spectral dependence of complex refractive indices is also ignored in the retrieval. The derived effective radius (R$_{eff}$),
25  volume concentration (V$_c$), the real (m$_R$) and imaginary (m$_I$) part of the refractive indices are summarized in Table 2. The errors of the retrieved parameters have been discussed in the relevant papers (Müller et al., 1999; Veselovskii et al., 2002; Pérez-Ramírez et al., 2013).

The retrieved particle size distributes in the range of 0.1 to 1.0 $\mu$m, with effective radius (volume-weighted sphere radius) of $0.33 \pm 0.10$ for both Palaiseau data and Lille data. The volume concentration is $15 \pm 5$ $\mu$m$^{-3}$cm$^3$ for Palaiseau data and $22 \pm 7$
30  $\mu$m$^{-3}$cm$^3$ for Lille data. The real part of the complex refractive indices retrieved from Lille and Palaiseau data are also in good agreement, giving $1.55 \pm 0.05$ and $1.52 \pm 0.05$ for the real part, and $0.028 \pm 0.014$ and $0.021 \pm 0.010$ for the imaginary part. The derived aerosol microphysical properties from Palaiseau and Lille data are consistent.

Number: 1       Author:     Subject: Cross-Out   Date: 17.08.2018 11:41:20

Number: 2       Author:     Subject: Inserted Text       Date: 17.08.2018 11:42:15

s

Number: 3       Author:     Subject: Cross-Out   Date: 17.08.2018 11:42:33

Number: 4       Author:     Subject: Sticky Note       Date: 17.08.2018 14:20:36

this was already all explained above

Number: 5       Author:     Subject: Comment on Text       Date: 17.08.2018 11:46:04

unit correct?

Number: 6       Author:     Date: 17.08.2018 11:47:37

Can you also report the resulting SSA?

[Figure]

**4.2.3 Direct radiative forcing effect**

The UTLS plumes observed on 24 and 28 August in Lille and Palaiseau are optically thick, with extinction coefficient about 10 times higher than in the volcanic ash observed by Ansmann et al. (1997) in April 1992, 10 months after the eruption of Mount Pinatubo. The radiative forcing imposed by the observed layers is a curious question. We input the retrieved microphysical properties into GARRLiC/GRASP to estimate the DRF effect of the UTLS plumes in Lille and Palaiseau. The calculation

5   includes the $0.2 - 4.0$ $\mu$m spectral range. We assume the vertical volume concentration of aerosols follows the extinction profile in Figure 8 and 9. The surface BRDF parameters for Lille and Palaiseau are taken from AERONET. The upward and downward flux/efficiencies, as well as the net DRF ($\Delta F$, with respect to a pure Rayleigh atmosphere) of the UTLS aerosol layers are calculated and Table 3 shows the daily averaged net DRF (W/m$^2$) at four levels: at the bottom of the atmosphere (BOA), below the UTLS layer, above the UTLS layer and at the top of the atmosphere (TOA). For the layer observed in Lille

10  on 24 August, the top and base of the UTLS are selected as: 18.4 km and 16.7 km and for Palaiseau observations, they are 20 km and 17.0 km.

At the top of the atmosphere, the net DRF flux is estimated to be -1.2 Wm$^{-2}$ and -3.5 Wm$^{-2}$ for Lille and Palaiseau data, respectively. The corresponding forcing efficiencies are -7.9 Wm$^{-2}\tau^{-1}$ and -21.5 Wm$^{-2}\tau^{-1}$ . At the bottom of the atmosphere, the net DRF flux is estimated to -12.3 Wm$^{-2}$ for Lille data and -14.5 Wm$^{-2}$ for Palaiseau data. The corresponding forcing

15  efficiencies are -79.6 Wm$^{-2}\tau^{-1}$ and -89.6 Wm$^{-2}\tau^{-1}$. We noticed that the difference in net DRF flux between the layer top and layer base is significant. For Lille data, we obtained 9.9 W/m$^2$ of difference between the top and the base of the UTLS layer and for Palaiseau, we obtained 11.1 W/m$^{-2}$. Because of the high imaginary part of refractive indices, the UTLS aerosols have the capacity of absorbing the incoming radiation, thus reducing the upward radiation at the top of the UTLS layer and the downward radiation at the base of the UTLS aerosol layer. The heating rate of the UTLS layer is estimated to be 3.3 K/day for

20  Palaiseau data and 3.7 K/day for Lille data. This qualitatively explains the increase of temperature within the UTLS layer, as observed by the radiosonde measurements shown in Figure 2.

**5   Discussion**

[Figure]
Based on the satellite measurements of UVAI, CO concentration and CALIPSO profiles, we can conclude that the UTLS layers observed over northern France are originated from Canadian smoke. The layers are transported by UTLS advection and are

25  characterized by high UVAI, CO concentration, together with enhanced depolarization ratios. The smoke particles were lofted to the UTLS probably by the pyro-cumulus clouds observed by GEOS-15 on 12 August 2017. The UTLS smoke plumes reported in this study have traveled for more than 10 days from the source region, considering that the smoke plumes were lofted on 12 August and then observed by the Lidar systems during 24–31 August. The high altitude of the smoke plumes prevented them from mixing with many other aerosol types during the transport. So the observed plumes are most likely to be aged smoke

30  particles.

The measurements revealed high depolarization ratios in the UTLS aerosol at 355 and 532 nm. In particular, the range of depolarization at 355 nm is $23 \pm 3\%$ to $28 \pm 4\%$, while at 532 nm it is about $19 \pm 3\%$. The depolarization ratio at 1064 nm is

Number: 1     Author:     Subject: Comment on Text     Date: 17.08.2018 14:22:27
Is this part of the discussion? It is more a summary from above.

Number: 2     Author:     Subject: Cross-Out  Date: 17.08.2018 12:51:46

Number: 3     Author:     Subject: Comment on Text     Date: 17.08.2018 14:22:37
do you have evidence for that?

Number: 4     Author:     Subject: Cross-Out  Date: 17.08.2018 12:53:08

Number: 5     Author:     Date: 06.08.2018 14:58:57
PARTICLE

[revised manuscript text omitted]

Number: 1        Author:    Subject: Comment on Text      Date: 17.08.2018 14:22:59
at which altitudes?

Number: 2        Author:    Subject: Comment on Text      Date: 17.08.2018 14:23:15
at which altitude? What is the difference between Burton 2015 and 2012? Please explain briefly.

Number: 3        Author:    Subject: Comment on Text      Date: 17.08.2018 12:56:31
how small?

Number: 4        Author:    Subject: Comment on Text      Date: 17.08.2018 12:57:44
? show evidence?

Number: 5        Author:    Subject: Comment on Text      Date: 17.08.2018 12:57:48

Number: 6        Author:    Date: 06.08.2018 15:07:27
of smoke particles from North- American fires?

Number: 7        Author:    Subject: Comment on Text      Date: 17.08.2018 13:04:08
your study?

Number: 8        Author:    Subject: Comment on Text      Date: 17.08.2018 14:41:25
How does the growth of the smoke work: coagulation or an increasing sulfate shell? this is not the same

[revised manuscript text omitted]

Number: 1    Author:    Subject: Comment on Text    Date: 17.08.2018 14:23:57
Do you have a reference for that statement? What is consistent?

Number: 2    Author:    Subject: Comment on Text    Date: 17.08.2018 14:24:15
for both parts? And do you have reference for your statement?

Number: 3    Author:    Subject: Comment on Text    Date: 17.08.2018 14:30:52
I believe, the same holds for depolarization.....

[Figure]

**6 Conclusion**

In 2017 Summer, large-scale wildfires spread in the west and north of Canada. In the mid-August, severe fire activities generated strong convections that lofted smoke plumes up to the UTLS. The smoke plumes were transported by advection in the UTLS and observed by the Lidar systems in northern France after long-range transport. Multiple satellite sensors recorded the long-range transport of the smoke plumes. The smoke plumes have undergone more than 10 days of aging process before being

5    observed by the Lidar systems in northern France. The optical and microphysical properties derived from Lidar observations revealed important features of aged smoke, including the optical depth, Ångström exponent, Lidar ratio, depolarization ratio, refractive indices, particle size and concentrations. The increase of smoke particle depolarization ratios versus transport time is first presented and points to smoke particle coagulation during the transport. The DRF estimation indicated that the UTLS aerosols strongly reduce the radiation reaching the bottom of the atmosphere by absorption. The derived heating rate of the

10   UTLS layer agreed with the temperature increase within the layer, according to the radiosonde measurements. This study provides a good reference for the characterization of long-transported smoke aerosols and shows the capability of multi-wavelength Raman Lidar in aerosol profiling and characterization.

*Data availability.* All the data used in this paper are available upon request to the corresponding author
[Figure]

*Acknowledgements.* We wish to thank ESA/IDEAS program who supported this work. FEDER/Region Hauts-de-France and CaPPA Labex

15   are acknowledged for their support for LILAS multiwavelength Raman LiDAR and MAMS system.
[Figure]
 2020-ACTRIS-2/LiCAL Calibration center, ACTRIS-France, ANRT France, CIMEL Electronique and Service National Observation PHOTONS/AERONET from CNRS-INSU are acknowledged for their support. The development of Lidar retrieval algorithms was supported by Russian Science Foundation (project 16-17-10241). The authors would like to acknowledge the use of GRASP inversion algorithm (http://www.grasp-open.com) in this work. The SIRTA observatory and supporting institutes are acknowledged for providing IPRAL data. Finally we thank all the co-authors for their

20   kind cooperation and professional help.

Number: 1     Author:     Date: 17.08.2018 14:42:38
Please upload the lidar data to EARLINET data base. AERONET data should be  available via aeronet web page, right?
I am also pretty sure your radio sounding data can be found somewhere. Furthermore please indicate how to access to all the satellite data you used. This is now demanded for all publications in ACP.

Number: 2     Author:     Date: 17.08.2018 14:26:07
shouldn't you acknowledge actris, earlinet, and nasa? are you sure you properly acknowledged all others with respect to the data you used?

Number: 3     Author:     Subject: Comment on Text     Date: 06.08.2018 15:19:44
grant numbers?

Number: 4     Author:     Date: 17.08.2018 14:42:52
author contributions....please include.

[revised manuscript text omitted]

Number: 1          Author:     Subject: Sticky Note          Date: 17.08.2018 14:27:25
In my opinion unnecessary.

[Figure]

[Figure]

**Figure 11.** The variation of particle linear deploarization ratio at 532 nm versus transport time. The transport time in this study and in Haarig et al. (2018) is estimated by assuming that the smoke plumes were emitted on 12 August 2017. The particle depolarization ratio (red solid circle) in Fromm et al. (2008) is calculated using the volume depolarization ratio and backscattering ratio measured on 24 June, 2001. The transport time for Burton et al. (2012) was estimated from HSPLIT back trajectories (Stein et al., 2015; Rolph et al., 2017), as it was not available in the paper. The other transport times are taken from the original publication.

---

## Author Comment (AC1) · 16 Nov 2018

Thanks for your helpful advise. Please find our point-by-point response in the supplement. Corrections have been made considering all the suggestions from the reviewers, please see our preliminary revised manuscript in the supplement. Some of important changes are highlighted in blue color.

Please also note the supplement to this comment:
https://www.atmos-chem-phys-discuss.net/acp-2018-655/acp-2018-655-AC1-supplement.zip

---

## Author Comment (AC3) · 16 Nov 2018

Thanks for your helpful advice. Corrections have been made considering your suggestions as well as other reviewers'. Please find our point-by-point response and first revised version in the supplement.

Please also note the supplement to this comment:
https://www.atmos-chem-phys-discuss.net/acp-2018-655/acp-2018-655-AC3-supplement.zip

2018.

---

## Author Comment (AC4) · 16 Nov 2018

Thank you very much, Albert ! Your advice is very helpful. We made corrections in the manuscript and also re-organized the paper. Please find our point-by-point response and first revised version of the manuscript in the supplement.

Please also note the supplement to this comment:
https://www.atmos-chem-phys-discuss.net/acp-2018-655/acp-2018-655-AC4-supplement.zip

2018.

---

## Author Response (AR1)

**Main changes are listed as below:**

1. The title is changed, as suggested by Albert Ansmann. 'Stratospheric' and 'lower stratosphere' are used to specify the altitude of the smoke layers in this study.

2. The abstract is rewritten

3. The introduction and conclusion are re-shaped.

4. Figure 11 in the discussion manuscript has been removed, and we also removed the argument about particle depolarization ratio increasing with transport time. More efforts are needed in investigating CALIOP data and their uncertainty before drawing convincing conclusions. This work is planned for the next step.

5. The error estimation part is moved in the Appendix

6. Discussion about the error of molecular scattering in the optical thickness is added

7. The methodology is now presented before the observations

8. The PLDR355 nm on 31 August is corrected to 0.28±0.08 (error recalculated)

9. More information about GARRLiC/GRASP algorithm is added.

10. Comments about the error of regularization inversion are discussed.

11. The error of Klett method is discussed. A comparison of backscatter coefficient between Klett and Raman is shown in the response file, showing that Klett can provide very consistent backscatter coefficient profile, which justifies the application of Klett method in this study.

12. Discussion section is reshaped and polished.

13. Table 1 is added to summarize the three Lidar systems

14. Figure 2 is added to illustrate the ascent of the smoke plume when exposed to sunlight

15. Smoke layer thickness, mean extinction coefficients are added in Table 2

16. Figure 10 in the discussion version is removed because materials are enough to show the origin of the smoke.

17. The unit of PLDR is changed from % to unitless, to avoid confusion with relative error.

18. 'Author contribution' is added and 'Data availability' is implemented.

This is a revised manuscript responding to all the four reviews.

The paragraphs in red in this manuscript are modifications corresponding to the response to the four reviews.

Different colors refer to different reviewers

And the numbers coincide with the ones in the response file, which was uploaded as supplement on 16 Nov 2018.

5 **Anonymous Referee #1:** AR1 A#. Page 35-37
**Anonymous Referee #2:** AR2 A#. Page 38-42
**Anonymous Referee #3:** AR3 A#. Page 43-49
**Albert Ansmann:** AA A#. Page 50-68

[revised manuscript text omitted]

**Reply To:** The introduction has been modified as suggested by ` AR3 supplement. `, ` AR3 A2. `, ` AR3 A3. `, ` AA A4. `, ` AA A11. `

**2 Methodology**

**Reply To:** The observations are put after the methodology, refer to AA A6. , AA A13, A16.

[revised manuscript text omitted]

10  (**P**hotomètre **L**éger **A**éroporté pour la **S**urveillance des **M**asses d'**A**ir, Karol et al. (2013)), capable to measure columnar aerosol optical depth (AOD) along the route. The configuration of the three Lidar systems is summarized in Table 1.

**Reply To:**  AA A5.

Figure 1 shows the normalized Lidar range-corrected signals and columnar AOD at 532 nm derived from sun photometer
15  measurements on 29 August 2017. The aerosol layers in the lower stratosphere, stretching from 16 to 20 km, were detected by the three Lidars. The IPRAL Lidar system in Palaiseau detected the aerosol layer in the range of 16–20 km on 29 August. The columnar AOD showed no significant variations, staying between 0.30 and 0.40, from 1000 UTC to 1600 UTC and started decreasing from 1700 UTC. Along the route Palaiseau-Lille, MAMS Lidar observed a layer between 16 and 20 km consisting of two well-separated layers. The columnar AOD was very stable, around 0.40, all along the route from Palaiseau to Lille.
20  Lidar LILAS in Lille observed a shallow layer between 18–20 km at about 0800 UTC on 29 August. The thickness of the layer increased to 4 km until 1600 UTC. The columnar AOD increased from 0.20 to 0.40 from 0800 UTC to 1400 UTC. The lidar quicklook indicated that the aerosol content in the lower troposphere did not show significant variations during 0800 UTC and 1200 UTC, so the increased optical depth, 0.2, came mainly from the contribution of the stratospheric aerosol layer.

Figure 2 shows the Lidar range corrected signal at 1064 nm on 24–25 August 2017. The plume between 17 and 18.5 km is the
25  smoke layer. Due to cirrus clouds and low clouds in the troposphere, the lidar signals in the plume are interrupted. In nighttime, the plume base is stable at about 17 km. Just starting from the sunrise time at 04:51 UTC, a gradual and obvious ascent is observed. In 3–4 hours, the plume base ascended about 0.6 km. Between 10:00–16:00 UTC, the plume base stayed stable. The ascent of smoke plume was also presented in Ansmann et al. (2018) and Khaykin et al. (2018). Khaykin et al. (2018) mentioned that the plume ascended very fast during the first few days after being injected into the troposphere. Based on the
30  observation in Figure 2, we derived the ascent rate of approximately 2.1–2.8 km per day, considering that the sunshine duration is 13 hours (according to the latitude of Lille site) and that the vertical speed of the plume is constant. Ansmann et al. (2018) explained that the ascent of the plume may be related to the absorption of soot-containing aerosols and the wind velocity in the stratosphere. Figure 2 shows that the plume does not continuously ascend in the daytime. One possible explanation we infer is that the self-heating and the wind shear reached an equilibrium point in the plume, so it moved neither upward nor downward.

**Reply To:** `AR1 A.` , `AR2 A.` , `AR3 A.` , `AA A.` , one figure of lidar observation is added, to illustrate the ascent of the plume.

[revised manuscript text omitted]

**Reply To:** AA **A19.** After re-calculation, we derived 0.28±0.08 for the molecular depolarization ratio at 355 nm.

[revised manuscript text omitted]

**Reply To:** The mean extinction coefficients are added in the table, refer to AA **A7.** , AA **A21.**

[Figure]

(a)                                       (b)

**Figure 9.** (a) Extinction and backscatter coefficient, (b) particle linear depolarization ratio (PLDR), the extinction-related Ångström exponent (EAE) and backscatter-related Ångström exponent (BAE) retrieved from LILAS observations between 2200 UTC, 24 August 2017 and 0030 UTC, 25 August 2017, Lille. The errors of extinction, backscatter coefficient and corresponding Ångström exponent at 355 and 532 nm are attributed to the error of the optical depth.

**Table 3.** Retrieved microphysical properties using the Lidar data in Lille and Palaiseau. Extinction and backscatter coefficients shown in Figure 9(a) and 10(a) are averaged in the range of 17–18.0 km and 17.5–19.5 km, respectively. The averaged extinction and backscatter coefficients are used as the input of regularization algorithm to retrieve particle microphysical properties.

| | $R_{eff}$ ($\mu m$) | $V_c$ ($\mu m^3 cm^{-3}$) | $m_R$ | $m_I$ |
|---|---|---|---|---|
| Lille, 24 August | $0.33 \pm 0.10$ | $22 \pm 8$ | $1.55 \pm 0.05$ | $0.028 \pm 0.014$ |
| Palaiseau, 28 August | $0.33 \pm 0.10$ | $15 \pm 5$ | $1.52 \pm 0.05$ | $0.021 \pm 0.011$ |

[Figure]

(a)                                           (b)

**Figure 10.** (a) Extinction and backscatter coefficient, (b) the particle linear depolarization ratio (PLDR) at 355 nm, the extinction-related Ångström exponent (EAE) and backscatter-related Ångström exponent (BAE) (between 355 nm and 532 nm) retrieved from IPRAL observations between 1920 and 2120 UTC, 28 August 2017, Palaiseau.

**Table 4.** Daily averaged net DRF flux calculated by GARRLiC/GRASP. Aerosol microphysical properties in Table 3 and aerosol vertical distributions in Figure 9(a) and 10(a) are used to calculate the DRF effect at the following four vertical levels.

| $\Delta F$ (W/m$^2$) | TOA | BOA | layer top | layer base |
|---|---|---|---|---|
| Lille, 24 August | -1.2 | -12.3 | -2.1 | -12.0 |
| Palaiseau, 28 August | -3.5 | -14.5 | -2.5 | -13.6 |

Thanks to the reviewer for his/her helpful advice, please find our answers below:

GENERAL COMMENT

The paper presents an interesting study on long-range transported smoke aerosols, originated in Canada wildfires, in the UTLS (Upper Troposphere/Lower Stratosphere) over Europe detected at several EARLINET stations in summer 2017 in combination with satellite observations. Optical depth at 532 nm from 0.05 to above 0.20 were detected, with very weak spectral dependence. Other particle microphysical properties like Lidar ratio and particle depolarization ratios suggest the presence of aged smoke likely with complicated morphology. The retrieved aerosol properties allowed the computation of the direct radiative forcing (DRF) effect originated in the UTLS aerosol layers and the radiative heating rates in this layers that are coherent with the observed radiosonde temperature profiles. The paper is worthy to be published in ACP having in mind that evidences the capabilities of a lidar network focused on tropospheric research in obtaining valuable information on the UTLS aerosols. For this purpose both the advanced instrumentation and the analytical tools used are crucial. The paper is well written and offers valuable information for the reader. Nevertheless the clarification of some points will enhance the quality of the paper.

PARTICULAR COMMENTS

In this work it is especially relevant to get information on the accuracy and uncertainties of the retrievals. The AOD retrievals and the lidar ratio retrievals are clearly related in the analysis procedure used. In this sense, the approach followed for the computation of the UTLS AOD and Lidar ratio with the fixed lidar systems is stated and details on the error propagation and discussion on the accuracy and uncertainty of the retrievals is presented. Nevertheless, some points require additional clarification.

1. Thus, concerning the discussion on the error propagation, in Page 7, I have a question: Are the authors assuming the absence of errors in the molecular part? Having in mind the impact of an accurate thermodynamic profile on this assumption I do not see any information on the thermal profile used. Furthermore, although the computation of the uncertainties is applied in the analysis sections, it would be worthy to include some quantitative information concerning the final uncertainties of the UTLS AOD and UTLS lidar ratio retrievals in the last paragraph in section 3.1.1.

**A1:** The temperature and pressure profiles are taken from radiosonde measurement in the closest stations: Trappes 20 km from Palaiseau and Beauvechain 120 km from Lille

station. Although the radiosonde stations are not exactly collocated with the lidar observations, we found the spatial variations are minor after exanimating the variability of the temperature and pressure profile in the two stations in August 2017. We think the errors resulting from molecular scattering are not significant, so it is not considered in the total error estimation of the optical depth.

The total error is calculated following Equation (4) and (5), and the calculation process is quite straightforward, we think that it is not necessary to present the calculation details in the paper.

2. In the case of the MAMS lidar retrievals there are additional limitations and the issue of accuracy and uncertainty is particularly relevant. Thus, in spite of the auxiliary use of its data, it is necessary to include additional discussion on the reduced accuracy of this retrieval.

**A2:** The limitation of MAMS lidar inversion is discussed in the 'Methodology' section before the MAMS results are presented. In the revised version, we mention when presenting MAMS results, that "the MAMS results are limited mainly by the difficulty of quantifying the errors resulting from the lidar signal at high altitude and the assumption of vertically constant lidar ratio".

3. Considering the uncertainty in the AOD retrievals the AOD spectral dependence will present a large uncertainty that requires additional discussion.

**A3:** The error of the Angstrom exponent Å is derived from the following equation:

$$(\Delta \text{Å})^2 = \left( \frac{1}{\log \left( \frac{\lambda 1}{\lambda 2} \right)} \right)^2 \left[ \left( \frac{\Delta \tau_{\lambda 1}}{\tau_{\lambda 1}} \right)^2 + \left( \frac{\Delta \tau_{\lambda 2}}{\tau_{\lambda 2}} \right)^2 \right]$$

Considering the error of the two selected cases, the error of optical depth is about 10%, the estimated error of the Angstrom exponent is about 0.3 (absolute value, unitless); and if the error of optical depth is 15%, the resulting error of Angstrom exponent is about 0.5(absolute value, unitless). Compared to the extinction coefficient, the backscatter coefficients we derived are more reliable because they are rather consistent with the results from Raman inversions, except the 1064 channel. The main error of regularization input comes from the error in the spectral AOD and the backscatter coefficient at 1064 nm.

The above information is reorganized and added into the revised version.

4. More details about the procedure used in GARRLIC for the computation of the

aerosol DRF and the UTLS, layer heating rates must be provided.

**A4:** More information has been added in the manuscript to describe the general strategy of GARRLiC /GRASP and the input parameters for the calculation procedure. The theories and methodology of GARRLiC/GRASP can hardly be well presented in a short section. So we suggest the readers to refer to previous publications about GARRLiC or GRASP. GRASP is an **open source algorithm**, anyone who is interested in using GRASP to reproduce the results in this paper or to invert their own measurements, is **very welcome** to download the algorithm here: https://www.grasp-open.com or contact us by email.

Minor changes

5. The references on the EARLINET network must include a recent reference that updates the features of the network: Pappalardo, G., Amodeo, A., Apituley, A., Comeron, A., Freudenthaler, V., Linné, H., Ansmann, A., Bösenberg, J., D'Amico, G., Mattis, I., Mona, L., Wandinger, U., Amiridis, V., Alados-Arboledas, L., Nicolae, D., and Wiegner, M.: EARLINET: towards an advanced sustainable European aerosol lidar network, Atmos. Meas. Tech., 7, 2389-2409, https://doi.org/10.5194/amt-7-2389-2014, 2014.

**A5:** It is added.

6. Please consider the following reformulation of the statement on Page 6 Line 2 "at this temperature clouds consist mainly of ice crystals" In order to increase the clarity of the test include the following changes in the first paragraph of section 3.1.1: Substitute:" The integral of the extinction coefficient over the UTLS layer, expressed below, is compared with the pre-calculated optical depth" by "The UTLS AOD is calculated by the integral of the extinction coefficient over the UTLS layer, expressed below". And after equation (2) reformulate the statements: "This pre-calculated optical depth is derived from the elastic channel at 355 and 532 nm. The method is widely used in cirrus clouds studies (Platt, 1973; Young, 1995). " as follows" by "This derived value of AOD is compared with the pre-calculated optical depth obtained from the elastic channel at 355 and 532 nm using a method widely used in cirrus clouds studies (Platt, 1973; Young, 1995)" In page 7 Line 7 change: "We calculate the signal mean within a window of 0.5 km..." by "We calculate the lidar signal mean within a window of 0.5 km. . ." In page 9 line 13 consider changing "intervals" by "periods".

**A6:** These statements are re-phrased; some are not exactly modified as suggested but in a similar manner.

Reply to Anonymous Referee #2:
Thanks a lot to the reviewer for his/her helpful advice. Please find our point-by-point response below.

General assessment and major comments

This study provides lidar measurements from two sites in Northern France, showing long range transported smoke in the UTLS. The absorbing nature of smoke is crucial for the stratospheric height ranges, concerning both heating rates (HR) and direct radiative forcing (DRF). The authors try to estimate the DRF and HR and their results show decrease of the radiation reaching the surface and an increased HR due to the absorption of the solar radiation at TOA. In general, I find this study very interesting and of high value. It is a study that fits well in the EARLINET special issue, since it demonstrates the value of EARLINET lidars for atmospheric research in both troposphere and stratosphere. However, before proceeding with publication in ACP, I strongly suggest that the authors would revise the following points:

1. Page 11, Lines 18-27: "The spheroid model was used to retrieved dust properties (Dubovik et al., 2006; Mishchenko et al., 1997; Veselovskii et al., 2010). But it is not clear if this model is applicable to soot particles with complicated morphology. The size of smoke particles is expected not too big so that we choose to apply regularization algorithm with sphere model." The retrieved microphysical properties seem to be associated with high uncertainties, since the shape used (spherical) does not reproduce the depolarization measurements and it should not reproduce accurately the backscattered light measurements either. The reported uncertainties in Table 2 refer to cases of spherical particles and are not representative. Please provide a better assessment of the retrieval uncertainties.

**A1:** It is true that spheres do not represent correctly smoke particles, neither spheroid. Our retrievals of dust particles demonstrated, that when spheres were used instead of spheroids, the algorithm was still able to provide reasonable estimates of volume and effective radius (Veselovskii et al, JGR 2010). The main errors were attributed to estimations of the refractive index. So we expect that in the case of smoke estimations of radius and volume are also possible.

2. Regarding the DRF calculations: these are based on the retrieved microphysical (point 1) properties which, as discussed above, are derived from the 3b+2a regularization inversion and are associated with (most probably) high

uncertainties. Especially for the imaginary part this uncertainty is expected to be the highest (Burton et al., 2016). Please provide a better assessment of the retrieved property uncertainties and quantify the uncertainties of the DRF calculations accordingly. If this is not possible, omit section 4.2.3 from the manuscript. This also applies to Page 14, Lines 9-13, where the derived complex refractive index is compared to other studies. Omitting 4.2.3 would not affect the quality of the paper, since the authors already provide important results on smoke optical properties and microphysical estimates.

**A2:** From our simulation studies we estimate errors of V (volume concentration) and $R_{eff}$ as 30%, for $m_R$ it is +-0.05 and $m_I$ 50%. These are typical values and we are not able to evaluate the effect of shape of on retrievals. But basing on dust studies, we expect it to be similar.

The deficiency of using sphere model is its not being able to reproduce the depolarization effect. However, the estimation of the radiative effect is not so sensitive to the depolarizing effect of the particles. Indeed, the uncertainty of the imaginary part of the complex refractive indices is much higher than the other parameters and it is strongly dependent on the shape of the particles, but the values we present are quite reasonable for previously reported absorbing smoke. We think the estimated radiative forcing is quite representative and the heating of smoke predicted by the DRF is able to explain the ascending trend of the plume, as shown in the newly added figure. Although the values suffer from some extent of uncertainties, we would like to keep section 4.2.3 and we will mention in the manuscript that the uncertainty of the retrieved aerosol microphysical properties affects the accuracy of the DRF estimation.

3. Another issue addressed in this study is the increase in particle depolarization ratio at 532 nm which is attributed to the particle aging. The authors gathered observations of the particle linear depolarization at 532nm from previous studies and have also included the results obtained from the present study. Nevertheless, the only visible trend seen in Figure 11 results from CALIPSO measurements. From the ground-based lidars in Lille and Palaiseau, there is no obvious increase at 532nm. In conclusion, the phrase "we found an increase in depolarization versus transport time" in the manuscript abstract should be changed to "CALIPSO observations of the UTLS smoke layer suggest an increase in depolarization at 532nm versus transport time".

**A3:** We agree that the main increasing trend of the depolarization is indicated by the CALIPSO measurements. However, the CALIPSO data are questionable because of the high noise level.  Moreover, the RH of the smoke plumes is not known. As a result, we cannot draw really convincing conclusion about the changes of depolarization ratio

during the aging process. At current stage, we decide to remove this part from the manuscript and more efforts will be made to investigate this issue and re-assess CALIPSO data.

Minor comments

4.  Page1, Line 9: "Typical particle depolarization" the meaning of the word typical should be clarified by the authors, meaning what is the definition of linear particle depolarization ratio used? (is it the cross/parallel ratio or the cross/total ratio?)

**A4:** After re-organizing the paper, the definition of particle linear depolarization ratio is in Section 2. The methodology is presented before the observation section, so this problem is avoided.

5.  Page 1, Line 10: "The relatively high depolarization ratios and such spectral dependence are an indication of a complicated morphology of aged smoke particles" The conclusion that the spectral dependence of the depolarization ratio is characteristic of aged smoke particles can be hardly drawn by two cases, i.e. the current one and the one reported in Burton et al. (2015). Please rephrase accordingly.

**A5:** This conclusion is drawn in Mishchenko et al., 2016

6.  Page2, Line 30: "We focus on the retrieval of the aerosol optical and microphysical properties from the Lidar measurements". The authors should highlight that the depolarization ratio values are not reproduced in the retrieval of the microphysical properties.

**A6:** Yes, this message is given in section 4.2.3 as the limitation of the retrieval.

7.  Page 4, Line 2: Please change the phrase "showed an increase of temperature in the stratospheric smoke layers" to "An increase of temperature due to the presence of smoke aerosols in this region" or something similar.

**A7:** Modification has been made in the revised manuscript.

8.  Page 5, Line 3: Change the phrase "A plume with relatively high UVAI first occurred over the British Columbia on 11 August, and the intensity of the plume was moderate" to "a plume of moderate intensity and relatively high UVAI, first occurred over British Columbia on 11 August. Page 5, Line 4: Please change the phrase "and the UVAI in the center of the plume reached above 10" to "and the

UVAI in the center of the plume reached above 10, as indicated by the grey area on the plot (Fig 4)"

**A8:** Modification has been made in the revised manuscript.

9. Page 6, Line 1: "We have examined the temperature profiles" Did you use radiosonde measurements? Please provide more info.

**A9:** Yes, it is radiosonde measurements from Wyoming radiosonde stations, data can be found here: http://weather.uwyo.edu/upperair/sounding.html

10. Page 6, Line 2: "the temperature drops below -38_C, at which temperature the cloud droplets mostly turn to ice phase" Please provide relevant reference.

**A10:** Please refer to Kârcher et al., 2003, A parameterization of cirrus cloud formation: Heterogeneous freezing.

11. Page 6, Line 8: "The increasing trend of the depolarization ratio is probably due to aerosol aging" As discussed above, this is a hardly drawn conclusion. Please rephrase accordingly.

**A11:** We decide to remove this argument.

12. Page 7, Line 7: "we can calculate the optical depth of the cirrus cloud" Please change "cirrus cloud" to "UTLS aerosol layer" since this is what you refer to in this case.

**A12:** Corrected.

13. Page 7, Line 12: change the phrase "are considered as the major error sources of the optical depth" to "are considered as the major error sources in the estimation of the optical depth"

**A13:** Corrected

14. Page 7, Line 13: "based on the statistical error of photon distributions" Please provide more info on the definition of the noise of your lidar measurements. Do you take into account the systematic errors?

**A14**: The error of the lidar signal is estimated based on the assumption that the photon-counting detection mode of the photomultiplier follows Poisson distribution. Then signal error is given by the covariance of the Poisson distribution. Systematic error of photon-counting detection is negligible especially in nighttime measurements, so it is not taken into account in the error estimation. We estimated about 3% of error for nighttime signal. In order to account for the interference of sunlight, we roughly use 5%

for the error in daytime.

15. Page 7, Line 22: change the phrase "of the error of optical depth" to "of the error of optical depth to the estimation of the Lidar ratio"

**A15:** corrected

16. Page 10, Line 15: typing error, change volume depolarization ratio at 355 nm to molecular depolarization ratio at 355 nm.

**A16:** Corrected

17. Page 14, Line 14: "Smoke in dry conditions have higher refractive indices than that in wet condition" Provide relevant reference.

**A17:** After reconsideration, we think this statement is not strict. Studies have shown that fresh smoke has a broad range of hygroscopicity. The study of the hygroscopicity of aged smoke is quite limited and requires more observational and experimental efforts. Additionally, the aging process could be very complicated considering possible effects related to the photochemical process, fuel types, particle coagulation, secondary aerosol generation and so on.

We decide to remove this comment and mention in the revised manuscript that "the hydroscopicity of aged smoke is not yet well revealed."

18. Figure 6: The x axes on CALIPSO plots should be the same in order to show the variation. Also the phrase in the caption "The profiles of backscatter coefficient and particle linear depolarization ratio (PLDR)" could be changed to "The profiles of backscatter coefficient and particle linear depolarization ratio (PLDR) at 532nm from CALIPSO" Figure 7: The points on this figure should be larger to be more visible. Also, it would be better if the colors of the points are different for the two lidar systems.

**A18:** Corrected.

Thanks a lot to the reviewer for his/her helpful advice. Please find our point-by-point reply below.

The paper provides unique measurements of an extreme event of smoke advection from Canada to Europe. Smoke could be observed within the Troposphere as well as in the Stratosphere. I consider the measurements and the resulting data as very valuable and of interest for the scientific community and the topic fits well in the scope of ACP.

However, in my opinion the paper tries to cover too much topics at once (optical properties of smoke, microphysical properties, source analysis, change of depolarization with aging time by using Calipso data, radiative transfer calculations, estimating of heating rates, temperature changes in the stratosphere etc.). Thus, the paper is partly confusing and for some of the topics the proper fundament needed for the conclusion drawn are missing.

I therefore recommend first to focus on your key expertise and present only the unique measurements, which are already shown and exploit as much as possible (optical and inversion results). Of course you should also include the satellite data for source analysis. However please focus on the lidar measurments in France. Then, in another paper(s), the other topics could be covered (i.e. radiative transfer calculations and heating rates etc.) Especially the discussion of the impact of the smoke plumes need much more detail and evidence, as for example it is not shown nor discussed that temperature increases in the stratosphere at a certain height are not caused by the complicated upper atmospheric circulation including tropopause foldings etc. I therefore recommend for the second paper to include some expertise of the upper atmospheric dynamics and probably some modelling to show and prove the impact of the smoke layer with its possible warming on the overall general condition of the UTLS region above Europe.

I also think that the use of Garrlic/Grasp radiative transfer code to obtain radiative impact must be explained much better. Currently, it is not understandable how the calculations are performed.

From the current version, I also doubt for the evidence of the increase in smoke depolarization with aging time. I think the use of Calipso data for such a study is again a topic of its own. Uncertainties in the Calipso retrieval (see comments in pdf) should be discussed. Even more important, other influencing factors as the relative humidity, the altitude of the smoke with respect to ground level but also tropopause should be investigated. Among others influencing factors (fire source, burning types).

I therefore recommend major revision with the recommendation written above. Find

other comments below and as pdf-comments in the supplement.

Paper structure:

1. -The abstract is too long. Please shorten.

**A1**: The abstract is rewritten after all the corrections.

2. -The introduction is in my opinion a loose sequence of different paragraphs and therefore not constructive. Please revise and make it more focusing on your topic. E.g.: have such events been reported earlier? Is this the first time? . . .

**A2:** Modifications have been made.

3. -Several facts concluded in the observations section are again raised in the discussion. I think you should shape your paper. Either describe your observations only, and then make a discussion in a separate section or make conclusions also in the observations section but then discuss only new issues in the discussion section.

**A3:** The abstract, introduction, discussion as well as the conclusion have been re-shaped.

Terminology

4. -I would recommend not to use "upper troposphere/lower stratosphere UTLS" as a standard the term. This historical term covers all altitudes between 5 and 30 km and thus does not make clear that a significant portion of the smoke you detected is above the local tropopause. Please feel free and state, that there is stratospheric smoke as well as smoke in the free troposphere. In most times you anyhow refer to stratospheric smoke with your statements. . .

**A4:** We decided to change "upper troposphere/lower stratosphere UTLS" to lower stratosphere.

Major scientific remarks:

5. -What about the uncertainties of Calipso. For example for the PLDR, the particle backscatter coeff. is needed which is in turn calculated with a Klett-like approach, i.e. a a-priori lidar ratio. Thus the questions arises, which aerosol type was classified by Caliop and which lidar ratio was used to obtain the PLDR and is this correct for your aerosol type of investigation.

**A5:** This is a very useful comment. In Figure 6, we find that the observed plumes are not well classified. 1) The classification provides scattered aerosol types, such as polluted dust, elevated smoke, dust and volcanic ash. 2) The derived aerosol type sometimes

oscillates profile by profile, and even adjacent profiles could be classified into different categories. Different aerosol types correspond to different lidar ratio assumptions, but Figure 6 shows the mean profiles of backscatter coefficient and PLDR over a small range (latitude ±0.01), without considering the impact of aerosol mis-classification. In this situation, the error of CALIPSO results is barely expectable.

We added in the manuscript that the plumes are not well classified in CALIPSO data processing and this decreases the accuracy of CALIPSO results. But we still keep Figure 6, in order to the transport of the smoke plume from Canada to Europe. In addition, we remove the argument about PLDR increasing with transport time, because of the unknown error level of CALIPSO PLDR product.

6. -Depolarization: The molecular depolarization ratio depends on your filters used in the lidar. Are the theoretical values you stated valid for your system? And can you neglect temperature effects? Which molecular depol ratio value did you use for the PLDR calculation? The measured one or the theoretical one?

**A6:** The theoretical value of molecular depolarization ratio (specific to the Cabannes line) is about 0.36%. However, due to the imperfection of the lidar optics (and ther factors), the measured depolarization ratio in the aerosol-free zone is higher than the theoretical value. Our interference filters in LILAS system well block the rotational Raman lines. Figure 1 below shows the rotational Raman lines and Cabannes lines for the laser wavelength at 532 nm (in standard atmosphere), as well as the transmission function of the interference filters. We have estimated the total molecular depolarization ratio to be 0.4%, including the Cabannes lines and rotational Raman lines and found that the impact of the included rotational Raman lines on the molecular depolarization ratio is quite minor. However, in the historical measurements, our system LILAS measured about 0.8—1.3% at 532 nm, 1.2—1.8% at 355 nm and 0.7—1.0% at 1064 nm. The depolarizing effect of the optics, the misalignment and the error in the calibration procedure are expected to be responsible for the error of measured molecular depolarization ratio. In the calculation of the aerosol particle linear depolarization ratio and its error, we use 0.4% for the molecular depolarization ratio. The error level of molecular depolarization may look a bit astonishing but, fortunately, the total error of the particle depolarization ratio is much less dependent on it when the aerosol is optically thick and depolarizing, like in the presented cases. In addition, we measured cirrus clouds below the stratospheric plumes on 24—25 August, the derived PLDRs are about 45%, without noticeable spectral dependence. The results are very consistent with previously reported PLDR of ice clouds and can be regarded as a verification of our measurements.

[Figure]

Figure 1. The molecular backscatter coefficient of rotational Raman lines and Cabannes line for the laser line at 532 nm. The calculation is made under a standard atmosphere. Only oxygen and nitrogen are considered as scatters in the atmosphere.

7. -Radiative transfer: The methodology explanation is too short. It is not reproducible how you performed the radiative transfer. Which parameters did you use as input? Which are constrained by the algorithm? and so on... thus, either you introduce a much more detailed description of this method and the paper gets longer or you shift these calculations to a second paper.

**A7**: We agree that the explanation about GARRLiC/GRASP is not enough for reproducing the forcing effect of the smoke plumes. More information has been added in the manuscript to describe the general strategy of GARRLiC /GRASP and the input parameters for the calculation procedure. The theories and methodology of GARRLiC/GRASP can hardly be well presented in a short section. So we suggest the readers to refer to previous publications about GARRLiC or GRASP. GRASP is an **open source algorithm**, anyone who is interested in using GRASP to reproduce the results in this paper or to invert their own measurements, is **very welcome** to download the algorithm here: https://www.grasp-open.com or contact us by email.

8. -Inversion: Is it useful to perform an inversion when having only 3 elastic signals? I mean lidar ratio is not an independent variable in your case...please discuss this!

**A8**: The lidar ratio is not a completely independent parameter, because we introduce an extra constraint, which is the optical depth of the smoke layer.  Indeed, assuming vertically constant lidar ratio is not a favorable way in the Raman lidar community,

because it looks not realistic in some cases. But do not forget that, the particle depolarization ratio is almost vertically constant in the smoke layer. It indicates that the particles are well mixed in the smoke layer. Based on this fact, we have confidence to say that the lidar ratio within the smoke layer will not show significant variations. To assure this hypothesis, we compared the backscatter coefficient calculated from Raman and Klett method. The comparison in the selected two cases is shown in Figure 2. One can see that the differences in the backscatter profile between the two methods are quite minor. It indicates that the backscatter coefficient we calculated is reliable and assuming a constant lidar ratio in the smoke layer is not far from the truth.

There are actually **5 input parameters** in regularization algorithm. As to the extinction profile, it fits the pre-calculated optical depth so the mean extinction in the smoke layer is also trustworthy.

[Figure]

Figure 2. The comparison of backscatter coefficient, (left) 19:20—21:20 UTC, 28 August 2017, Palaiseau and (right) 20:30—00:30 UTC, 24—25 August 2017, Lille

9. What justifies using a sphere model when particle size is small.

**A9**: It is maybe a bit miss leading to say that the small particle size justifies the applicability of sphere model. What we really wanted to address is: the difference between spheroid scattering and sphere scattering is very minor when the particle size is small, and to not complicate the situation, we chose to use a simpler model, the sphere model.

The sensitivity of scattering of particles to the shape (spheres or spheroids) can be found in "*Dubovik, O., et al. (2006), Application of spheroid models to account for aerosol particle nonsphericity in remote sensing of desert dust*". Figure 3 is taken from

the Figure 26(b) in Dubovik et al. 2006.  It plots the lidar ratio at 532 nm as a function of the aerosol Angstrom exponent, which is an indicator of the particle size.

It can be seen that when the Angstrom exponent gets bigger, the difference of lidar ratio between spheroid and sphere model gets smaller.

[Figure]

Figure 3. Lidar ratio plotted as function of Angstrom exponent. Rv is the mean radius of the size distribution. Each line shows the dependence of the lidar ratio for aerosol defined by a lognormal bimodal distribution of spherical (red labels) fine mode (with median radii 0.12, 0.14 or 0.2 mm) and coarse spherical (red labels) or spheroid mode (blue labels) (with median radii 1.0, 2.0, 3.0 or 5.0 mm).

10. -Increase in stratospheric temperature. Your explanations are not convincing concerning the temperature increase. Did you consider also all other processes in the UTLS? May this only be normal variability? How long do you observed a temperature increase? So please add more detail or shift to another paper (I think this is a topic alone).

**A10**: Other process, for example the variation of ozone concentration can also result in temperature changes in the stratosphere. We investigated the temperature profile measured by radiosonde at Trappes (close to Palaiseau, France) in the last two weeks of August 2017. Figure 4(a) shows that the temperature in the stratosphere has obvious variations in August 2017. The region in the magenta box is distinct with others, because it is an obvious local maximum. Moreover, the spatial-temporal occurrence of this temperature peak coincides with the occurrence of the smoke plume. The same spatial-temporal coincidence appears to Beauvechain temperature observation (close to Lille station). Based on the observations in two independent observation site, we are confident that temperature increase in the plume layers is caused by the presence of the absorbing smoke layers instead of other reasons.

[Figure]

Figure 4. (a) Temperature at Trappes in August 2017. (b) Temperature profiles at Trappes and Beauvechain.

11. -Depolarization ratio vs. aging time: The provided graphic and literature review does not convince me of the given causality, please also investigate the RH, the height etc. vs depolarization ratio.

**A11**: We agree that CALIPSO data are very noisy from which we can hardly draw convincing conclusion about the depolarization increasing with aging time. The error level of CALIPSO measurements is also questionable regarding the stratospheric smoke layers. In addition, the RH of the smoke plume is not available in CALIPSO measurements and some ground-based lidar observations. As a result, we decide to remove the questionable argument about depolarization increasing with aging time. We need more investigations before getting more convincing results.

- Please also note the supplement to this comment: https://www.atmos-chem-phys-discuss.net/acp-2018-655/acp-2018-655-RC3- supplement.pdf

**The** modifications as below have been made following the comments in the supplement:

1. The abstract, introduction discussion and summary are reshaped.
2. Typing and grammatical errors, ambiguous statements and other minor errors pointed out by the reviewer have been corrected.
3. A table has been added to summarize the configuration of the three Lidars

4. Data will be uploaded to EARLiNET database after the final review session. More

information about data availability is added.

5. Acknowledgement is modified and author contribution is included.

6. Errors of Rayleigh scattering?

**AA6:** We use temperature and pressure profiles from the closest radiosonde profiles. The Rayleigh fit in the aerosol free zone is excellent so the errors resulting from molecular scattering are ignored in the error estimation.

7. Why not simply use the lidar ratio you retrieved with the other method or the ones reported by Haarig et al.? I consider using the total AOD as a constrained much more critical as you have to use one lidar ratio for all heights...you could also use the lidar ratio of the PBL obtained with LILAS or IPRAl for the MAMS Klett retrieval in the PBL.

**AA7:** MAMS system performed measurements between Palaiseau and Lille. There are three data points from MAMS, two of them are not collocated with IPRAL or LILAS, and only the third data point was obtained in Lille. We lack the information of the tropospheric aerosol along the road. Moreover, considering the noise in MAMS lidar daytime measurements, the AOD measurement collocated with the MAMS Lidar should be a more solid constraint. So we chose to process MAMS lidar measurements with an extra constraint of AOD.

8. Divide the paper into two papers

**AA8**: After re-shaping the paper and considering the advise from the other reviewers, we would like to keep the observation, inversion and radiative forcing estimation in the same paper. But, as you suggested, we added more information about GARRLiC/GRASP algorithm. We cannot present all the details about GARRLiC/GRASP because it is an integrated algorithm containing many modules. In the revised version, the basic strategies and the input parameters are introduced in more detail, so if the users want to reproduce the results, they can download this open-source algorithm and follow the instructions. The information we present in the paper is to show the general strategy of the algorithm.

Thanks a lot for this very detailed and helpful review. Please find our point-by-point answers below:

General remark:

The paper is well written and focuses on an interesting atmospheric topic: Long-range transport of pyro-cumulonimbus-related Canadian fire smoke in the stratosphere observed over France in August 2017.

An original and unique measurement strategy is selected: Two lidars at Palaiseau and Lille and one mobile system that measured the smoke during a travel from Palaiseau to Lille.

The highlight is the retrieval of the spectrum of the particle linear depolarization ratio measured at three wavelengths. There is a strong decrease of the depolarization ratio from 25% in the UV down to less than 5% in the near IR.

However, the discussion of the results is partly confusing and must be improved. Major revisions are necessary. Detailed comments:

1. *The title is misleading. The focus is on lidar observations (in France) of aerosol layers several kilometers above the tropopause, and not below the tropopause (upper troposphere). So: ... smoke aerosols in the lower stratosphere .... would be correct. Furthermore, the title does not indicate where you made these observations: .... Long range transport of Canadian fires smoke towards Europe observed over northern France ... , and not, for example, in eastern Asia or in the Arctic. Please improve!*

**A1:** We agree that the plumes shown in this paper are all in the lower stratosphere. We thought that 'Upper troposphere and Lower stratosphere' could be more recapitulative because of the variation of the position of the tropopause.

Considering this suggestion, we changed the title to **'Long-range transported Canadian smoke plumes observed in the lower stratosphere over northern France'**.

2. *P1: The abstract must be updated after the requested changes.*

**A2:** it has been updated in the revised version

3. *P2, L2: All abbreviations must be explained when they are mentioned for the first time (in the main text). The abstract is a stand-alone text, and does not count in this respect.*

**A3:** it has been corrected in the revised version

4. *P2, L25. After the foregoing paragraph, a paragraph is missing in which the literature is reviewed which is already available regarding the Canadian smoke period in August 2017: Khaykin et al. (GRL, 2018), Ansmann et al. (ACP, 2018), Haarig et al. (ACP, 2018). If there are more, please mention them as well. Such a literature of foregoing work is (always) required in the Introduction. What is already done and thus known? What is our new contribution?*

**A4:** it has been done in the revised version

5. *P3, L1.... : Again, please explain: LILAS, IPRAL, PLASMA, MAMS etc...*

**A5:** it has been done in the revised version

6. *It is quite confusing that Section 2, that contains first observations and in-depth analysis of all the satellite observations of the smoke including CALIOP (space lidar) measurements, .... is then interrupted by a 'dry' lidar methodology section. It would be better to have the technical*

*section first (as section 2, or as an Appendix) and then all the observations and discussions in the follow-up sections, continuously in sections 3, 4, 5.... without a break...*

**A6:** We agree that the 3rd section about methodology and calculation looks like an interrupt, so it has been moved to the 2nd section.

7. *P3, L28: With the AOD of the smoke layer and the vertical extent of the layer (from the lidar observation), the layer mean particle extinction coefficient can be determined. These values should be presented as well.*

**A7:** The mean extinction coefficients have been added into the table.

8. *P4, L13: Please explain EOS AM-1*

**A8:** EOS means 'Earth Observing System'. AM and PM mean the time when the satellite passes over the equator is in the morning or in the afternoon. This information has been deleted in the revised version because it is not so relevant to this study.

9. P4, Section 2.4: Isn't that a similar study of OMPS UVAI maps as already presented by Khaykin et al. (2018, in the supplementary part)? Should be mentioned.

**A9:** Yes, it is the same results with by Khaykin et al. 2018, products from the OMPS aerosol index are used to trace the evolution of the smoke plumes. Comments have been added in the text.

10. P5, L3: I see a clear and strong jump in the aerosol load from 11 to 13 August 2017 over the northern parts of western North America (Canada), in Figure 4. This is a clear and convincing indication that the pyrocumulonimbus cluster activities on 12 August were most probably responsible for these aerosol features. But this not explicitly written.

**A10:** Comments have been added in the text.

**11.** P5, L10: explain AIRS!

**A11:** AIRS means Atmospheric Infrared Sounder, this has been explained in the introduction part.

**12.** *P5, L10-25: The same holds for Figure 5, a clear and sudden increase in the CO concentration from 11 to 13 August is visible. Furthermore, this 12 August event seems to be the reason for the extreme smoke load over Europe on 21-22 August (Figure 5f) as reported by Ansmann et al. (2018). This should be also mentioned.*

**A12:** Comments have been added to explain the increase of CO concentration from 11 to 13 August.

**13.** P5, CALIPSO section 2.6: You already discuss the linear depolarization ratio, but the definition and explanations are provided later (in section 3). This is one of the reasons, why you better start with a more technical section right after the introduction.

**A13:** The technical section has been put just after the introduction.

**14.** P6, L3: Mixed-phase clouds can produce depolarization ratios from almost zero (mainly drops) to 40% (mainly ice crystals), so why should they show depol values of 26-35% only?

**A14:** We agree that the depolarization ratio of ice clouds is typically about 40%, and mixed-phase clouds in principle should possess a depolarization ratio ranging from a few percent to below 40%. This value, '26-35%', comes from the study of Sivakumar et al., 2003 which is based on the Sassen, 1995.  It is based on kind of specific

observational data but does not cover the large variability of mixed phase clouds. We agree to use particle linear depolarization ratio of a few percent to 40% for mixed-phase clouds and above 40% for ice clouds.

**15.** P6, L8: Why does an increasing trend in the depolarization ratio indicate aging of smoke particles? First of all, I do not see a trend in the noisy CALIPSO measurements. And second, the strong depolarization can be simply explained by the irregular shape of the smoke particles..., and why should the irregular shape change with time in the stratosphere where dry soot particles seem to dominate.

*A15:* Indeed, CALIPSO data are very noisy and some observed plumes in Figure 6 are not correctly classified, thus introducing unknown errors to the PLDR. We decide to remove the argument about PLDR increasing with transport time.

**16.** P6, section 3: Interrupt! A complex section is given, which is only interesting for lidar experts. But the paper is written for the research community dealing the atmospheric smoke and aerosols.

*A16:* This section has been put just after the introduction. In our opinion, it is necessary to present the data processing procedure and the error calculation. We adapted the standard Klett method to avoid an unrealistic assumption of lidar ratio. And this method should be presented to the readers so the results could be reproduced by anyone that is interested in.

**17.** *In Sect. 2.1, the lidars are explained (Raman lidars). And in Section3, we learn that you make only use of the Klett method. This makes quite a significant difference. The lidar ratios at 355 and 532 nm are not directly measured, you need to assume that the lidar ratio is height constant (and equal to the obtained layer mean value), so the solutions of the backscatter and extinction*

*coefficients at a given wavelength are not independent of each other, the profiles are rather similar. And at the end, you use the non-independent extinction and backscatter solutions in the lidar inversion retrieval to obtain the microcphysical properties. This inversion algorithm however needs independently measured backscatter and extinction coefficients at 355, 532 and 1064 nm (backscatter). Please mention and explain this 'contradiction' in more detail. What are the consequences for the inversion uncertainties?*

**A17:** Our observations during this event do not provide optimal data for retrieving extinction coefficient by Raman method. Because 1) for some cases where the plumes' extinction was high (>0.1 km-1), the measurements were made in daytime or with cirrus clouds below the smoke layers; 2) and the other cases where nighttime measurements were made without cirrus contamination, the extinction coefficients of the plumes are quite low. The calculation of extinction coefficient using Raman requires smoothing. The size of the smoothing window and the choice of smoothing method (least-square method is commonly used) is dependent on the Raman signal quality and will certainly impact the retrieved extinction coefficient. The calculation of the backscatter coefficient using Raman method is expected to be less dependent on the smoothing of Raman signal. Indeed, Raman method provide independent calculation of extinction and backscatter, but the smoothing applied on the extinction could introduce some artifacts or 'faked vertical variations' into the lidar ratio profile.

The lidar ratio is a quantity determined by the aerosol type and independent of aerosol concentration, like the particle linear depolarization ratio (PLDR). From the case study in our paper, it can be seen that the PLDR does not show noticeable variations versus altitude in smoke plume in the stratosphere. Neither does the backscatter Angstrom exponent. This is an indication that the smoke particles are well mixed and the properties of the particles are vertically homogeneous within the plume. So we expect that the lidar ratio in the smoke layer do not have strong vertical variations.

Moreover, we apply Klett inversion in the range of (layer_base – 1km) to (layer_top + 1km), to reduce the impact of the assumption of vertically constant lidar ratio.

In addition, we have compared the backscatter profile calculated using Raman method and Klett method, and the differences are quite minor. Please see Figure 2 in this document.

In the revised manuscript, we mention that the backscatter profiles calculated from Klett have been compared with the ones from Raman method (not shown) and they are very consistent. And due the assumption of vertically constant Lidar ratio, the profiles of extinction and backscatter coefficient are similar in the shape.

> **18.** *P7, L7: You calculate the signal means (within 500 m thick layers) at aerosol layer top and base. Please be precise. You probably use signals below the base and above the top only? What happens if there are traces of smoke above and below the layer, what uncertainty causes the use of noisy Rayleigh signals above and below the aerosol layer in the lidar ratio retrieval? All in all, be using the simple Klett method the uncertainty in the lidar ratio values (at 355 and 532 nm) must be at least 30%. All this needs to be clearly stated. We need overall uncertainties in the presented lidar ratios at the end of section 3.1.1.*

**A18**: We have considered the possible impact of smoke remnants when processing the data. To avoid including any smoke remnants in the base and top window, the two windows are selected in the region that does not touch the base and top of the layer. The Rayleigh fit in the region where the two windows locate is good (in daytime Rayleigh, signal in the top window is inevitably more noisy than in the night), so we are confident that the impact of smoke remnants on Klett inversion is reduced as much as we can do. In the manuscript, we add this sentence: "the window of layer top and base is chosen in a region where lidar signal fits well with molecular scattering profile."

**19.** The same for section 3.1.2: We need overall uncertainties in the depolarization values at the end of the section. If you have already 10% uncertainty in the volume depolarization ratio at 355 and 532 nm, the uncertainties in the particle linear depolarization ratios will be close to 20% for the particle depolarization ratio (especially at 355 nm). How trustworthy are such very large 355 nm particle depolarization ratios of 28% you present later on in the article? Even in the case of desert dust (rather irregularly shaped particles) the particle depolarization ratio is usually 25% or less. Please comment on this.

**A19:** The calculation of the overall error of PLDR follows this equation:

$$\left(\frac{\Delta\delta_p}{\delta_p}\right)^2 = F_R\left(\frac{\Delta R}{R}\right)^2 + F_{\delta_v}\left(\frac{\Delta\delta_v}{\delta_v}\right)^2 + F_{\delta_m}\left(\frac{\Delta\delta_m}{\delta_m}\right)^2$$

(1)

where R, $\delta_v$ and $\delta_m$ represent the backscattering ratio, volume linear depolarization ratio (VLDR) and molecular depolarization ratio. The calculation indicates that the error of the PLDR is mostly determined by the error of the VLDR, especially when R is larger than 10. As shown in Figure below

[Figure]

Figure 1. PLDR versus backscatter ratio R

At 355 nm, R is about 3 to 8, considering all the observations we have analyzed. At 532 nm, R is much higher so the error of PLDR depends mostly on the error of VLDR.

For the case where PLDR355 is 28% :

R $\approx$ 3.5,  $\triangle$ R=20%

$\delta_v \approx$ 0.15,  $\triangle \delta_v$ = 10%

$\delta_m \approx$ 0.004,  $\triangle \delta_v$ = 300%

The coefficients of the three terms in Equation (1) are:

$F_R$=2.28E-01,

$F_V$=1.15E+00,

$F_m$= 9.31E-06

And the overall error of the PLDR is about 14.38%.

The calculation process is written in a Matlab script and can be found here:

https://www.dropbox.com/s/xjxdg6kxoirb50b/error_depol.m?dl=0

However, we admit that using 15% for the error of PLDR at all the three wavelengths is not quite apposite. Because it is based on the assumption that the error of R is about 20%, but the error of R at 355 nm in the night of 31 August to 01 September is apparently higher than 20%.  (15% for PLDR532 is safe enough). We recalculate the error of PLDR355 for the observations  on 31 August, with

R $\approx$ 3.5,  $\triangle$ R=30%

$\delta_v \approx$ 0.20,  $\triangle \delta_v$ = 10%

$\delta_m \approx$ 0.004,  $\triangle \delta_v$ = 300%

we obtain about **19%** for the total error of PLDR 355, and by changing $\triangle$ R to 50%, we obtain **28%. Thus, we decide to modify the particle depolarization ratio at 355 nm from 28 ±4 % to 28 ±8 % .**

**P.S.  The error of PLDR532 is still below 15% in this case.**

20. *P9, Sect. 4: First question: The given standard deviations (in the text and in Table 1) indicate the uncertainty or the atmospheric*

*variability (in time and in the vertical profile) or the uncertainty in the retrieval? It is not clear to me. Please state that clearly. Table 1: I miss the information about the height (base, top) of the smoke layers in the stratosphere in the table. I would recommend providing the layer-mean extinction coefficients as well.*

**A20:** The quantity appearing after the symbol '±' is the uncertainty. Comments have been added in the text to clarify this quantity. And, the mean extinction coefficients of the smoke layer have been added in Table 1.

21. *P10, L15: If you have an uncertainty of 20% in the 355 nm backscatter value and 10% in the volume depolarization ratio then it is hard to get an uncertainty of 15% in the particle depol ratio. I ask this to force you to re-check the results concerning the very large 355 nm particle depolarization ratios of 28%.*

**A21:** It has been corrected to 28 ±8 %. Detailed explanations are presented in **A19**.

22. P10, L29, Figure 8 and P11, L5: The only reliable profiles are the Klett solutions for the backscatter profiles. The extinction profiles and the EAE profiles are estimates and are strongly based on the used height-constant (layer mean) lidar ratios. It should be made very clear that the solutions of backscatter and extinction are so similar (or better identical in the profile characteristics) because of the use of the Klett method and the assumption of a height-constant lidar ratio. In the discussion of the Angstrom values (AE) you may know that EAE = BAE +LRAE, as given in Ansmann et al. (JGR, 2002 on ACE-2 observations in Portugal). And that fits very well in your observational cases. However, if the assumed LRAE is height constant, and the BAE is height constant, EAE can only be height constant. But in reality, EAE and LRAE may vary. This should be mentioned. Without the Raman lidar approach you cannot provide information on the

'real world' EAE, extinction, and lidar ratio profiles. They are based on an assumption, and not on measured facts. So it is misleading to show profiles of extinction and EAE without saying that these are just retrieval products heavily controlled by the assumption of a height constant lidar ratio.

**A22**: As explained in A17, we chose to not use Raman inversion to avoid the distortion of the extinction profile brought by the smoothing. Considering that the PLDRs of the smoke layer do not show noticeable vertical variation (PLDR calculated using the backscatter ratio resulted from Raman inversion, instead of Klett), the particles in the smoke plumes are very likely well mixed so do not have significant variations in the concentration-independent quantities such as PLDR, LR and BAE…

We compared the profiles of backscatter coefficient from Raman inversion and Klett inversion in the two cases presented in the paper, as shown in Figure (2). It can be seen that the backscatter coefficient from Raman and Klett inversion are rather comparable.

It is true that the similarity of the extinction and backscatter coefficient profile is due to the assumption of vertically constant lidar ratio. However, we can be confident that, due to the homogeneity of the smoke layer, the lidar ratio literally does not have significant vertical variations and our assumption is valid.

[Figure]

Figure 2. The comparison of backscatter coefficient, (left) 19:20—21:20 UTC, 28 August 2017, Palaiseau and (right) 20:30—00:30 UTC, 24—25 August 2017, Lille

23. *P10, L30: 'The UTLS aerosol layer was between 17 and 18 km'. This is one of the sentences that forced me to ask: Why do you call that an UTLS layers. The layer is clearly in the lower stratosphere.*

**A23**: Yes it is in the stratosphere. We will correct it to 'lower stratosphere' to be more precise.

24. *P11, L13: Figure 10 is questionable and probably not representative to explain the long range transport including transport times. The CALIPSO Figure 6 tells us that the smoke layers were mostly at heights of 14-17 km over North America and the Atlantic, sometimes even below 14 km. And at these heights the wind speeds were much higher than at 17-19 km (in Figure 10). We need more trajectories! Especially for the CALIPSO heights (12-17 km) before we can make conclusions on the travel duration (in the discussion section 5). And one effect makes the use of trajectories difficult: The ascent of the layers from day to day by absorption of sunlight! This is not considered in any trajectory modeling. So trajectories are of limited use here. But 20 days of travel from Canada to Europe? That would be quite new, in view all the papers on Canadian smoke and long range transport. Even Khaykin et al. (2018) writes that the smoke needed 21 days to travel around the globe (at midlatitudes) to be back over Canada again, and that the smoke needed only 10 days or less to reach Europe.*

**A24:** It is true that the time between the first occurrence of the smoke plume generated by pyro-Cbs (12 August) in Canada and our first observations in France (19 August) is approximately about 10 days. However, as we can see in the UVAI maps from OMPS, the smoke plumes were distributed over very large areas and they are transported along

different pathways, moreover the wind fields in different areas are different. So it is not surprising that the plumes transported along different pathways have different transport time and that ~10-day is very likely the fastest transport time.

We can see, from the UVAI map, that on 18 August, the front of the smoke plume has arrived in Europe but the main bulk of smoke plumes was sustained in the east of Canada, where no fire spots were detected. We think that the smoke plumes in this whole event (refer to our observation from 19 August to 02 September) were mostly generated during 11--12 August to 15 August. The smoke plumes distributed over large area and they were then transported to Europe with different transport time.

**25.** P11, section 4.2.2: Again the question: How trustworthy are the solutions of the inversion method when only based on Klett optical properties, so that the basic information are backscatter profiles? Please comment on that.

**A25**: Summarizing aforementioned information, the backscatter coefficients from Klett method are rather consistent with from Raman. The The mean extinction coefficients are derived from transmission of the plume, which is also robust and does not suffer from vertical smoothing. We consider the optical data as very trustworthy as input of the inversion. The main error source of the inversion is the use of sphere model, especially for the imaginary part of the complex refractive indices. From our simulation studies we estimate errors of V (volume concentration) and $R_{eff}$ as ±30%, for $m_R$ it is ±0.05 and $m_I$ ±50%. These are typical values and we are not able to evaluate the effect of shape of on retrievals.

Discussion section:

**26.** P12, L26-28: I have my doubts that one singular event (the major pyrocumulonimbus cluster on 12 August) leads to so many smoke layer that you observed between 24-31 August. To my opinion, layers observed 10 days later maybe linked to the 12 August event, and maybe even layers observed on 24 August. But all other smoke layers later on were most probably triggered by other reasons and causes. And note that the absorbing smoke layers ascended during the travel (because of strong absorption of solar radiation). Khaykin et al. (2018) mentioned, 2-3 km per day during the first days. So layers can easily reach the lower stratosphere after 3 days when initially injected to 8 km height...

**A26**: As mentioned in **A24**, 10-day is probably the fastest transport time. Due to the large distribution of the plumes and the differences in the wind fields, the plumes are likely to undergo longer transport time.

Let's sort out the time series between 11 and 29 August:

- Emission process: the fire sources (indicated by the fire and thermal anomalies from MODIS) of the smoke plumes are mostly in the Northwest Territories, the Alberta and the British Columbia, large amount of smoke accumulated over the northwest of Canada and propagated easterly. After 15 August, the intensity of the fire activities reduced and the plumes have been transported away from the source region. And on 15—28 August, there were no new plumes with comparable quantity and scale, being generated in the areas with fire spots any more, indicating that the fire activities were weak during this period. We consider that the plumes observed in Europe were mostly generated before 15 August.

- Transport process: the plume that first arrived in Europe came from the 'tail' (see UVAI on 16 August), locating between Canada and the US, of the large bulk of the plume, while the main bulk of the plume still stayed in the north-east of Canada. The main bulk of plume left Canada on 19 August and then propagated to Europe. We can see that on 21 August, the intense plume over central Atlantic has lost 'connection' with the plume over Canada. In the following days, this

intense plume continued propagating to Europe and the transport speed over Europe is much slower. For example, during 25—27 August, the plume hovered over France was changing very slowly. Even on 28, 29 August, the plume changed its shape and intensity, we did not see any new smoke coming from the Canada.

- In addition, the transport time from Canada to Europe and the aging time (between the generation and the observation of the smoke plume) are not the same variable, because of the complicated spatial distribution of the smoke plume. The aging time can be longer than the transport time from Canada to Europe.

As to the ascending trend of the smoke plume, indeed, the plumes tend to ascend when absorbing solar radiation. But we expect that the vertical motion of the wind field should dominate the descending or ascending of the plume.

27. P13, L1: In the discussion of the very large 355 nm particle depol values, please check: There is a corrigendum note to the Burton 2015 paper (you will find it on the ACP page). In this corrigendum note it is stated that the 355 nm depolarization ratios were only 20.5% for the smoke case (and not 24% as erroneously obtained in the beginning of the data analysis and published in Burton et al. (2015) paper). So, again, the 28% you get at 355 nm are really 'outstanding' and must be re-checked.... Find out, for example, how large the impact of the Klett backscatter profile is in the retrieval of the particle depol ratio at 355 nm... by using plus/minus 10% backscatter coefficient profiles.

**A27**: Please see **A19.**

28. P13, L5-L22: You provide the suggestion (Figure 11) that the depolarization ratio increases with travel time. This is surprising,

because Nisantzi et al. (ACP, 2014, Injection of mineral dust into the free troposphere during fire events observed with polarization lidar at Limassol, Cyprus) find the exact opposite and show that in a figure: Decreasing depolarization ratio with transport time. However, their study is exclusively based on tropospheric smoke. And you combine tropospheric and stratospheric observations in your analysis.

Who is right? Maybe both! But first of all, one has to clearly distinguish aerosols in the troposphere and in the stratosphere. As mentioned, tropospheric as well as stratospheric smoke depolarization ratios are considered in your Figure 11. To my opinion, in this way you compare apples and oranges, and therefore the conclusions maybe wrong, at least are 'dangerous'. In the troposphere, coagulation and interaction with gases and moisture can take place, as you discuss so that aging leads to growing particles. According to measurements they get coated (Dahlkoetter et al, ACP 2014), they get coated with liquid stuff, so they get more and more spherical. And that influences the depolarization ratio. The depol ratio decreases, in accordance with the Haarig et al. (2018) results. In the stratosphere all this is less probable, and coating of soot particle is practically suppressed because the moisture and all the gases are not given in the stratosphere. It seems to be that the particles in the stratosphere remain unchanged during the travel. That is what I see also from the noisy CALIPSO observations in Figure 6 and what is also shown in the Haarig paper. But maybe I am wrong. Please provide us with a convincing explanation why aging leads to an increase in the depolarization ratio!

Disregarding my personal opinion, as a consequence of the statements above and the rather different particle aging processes in the troposphere and stratosphere...: If you want to show Figure 11 than you need to clearly indicate the stratospheric values, may be by very different symbols, e.g., by big open circles. We need a strong contrast between

tropospheric and stratospheric values... The lower values in the Figure are obviously tropospheric depol values (apples) , and the higher ones are stratospheric values (oranges) . Furthermore I have my doubts about transport times of more than 14 days.... The related trajectories are always very uncertain and thus questionable.

Final point: I think the primary goal of the discussion section should focus on the comparison with other findings, especially with Haarig et al. (2018), and also Burton et al. (2015, upper tropospheric smoke case). As mentioned several times, your results are based (to a large fraction) on the use of the Klett method. And this includes the determination of the particle depolarization ratios because the Klett backscatter profiles are needed. This can be a significant reason for deviations, especially in the case of the products at 355nm.

**A28**: Nisantzi et al, 2014 reported a case of smoke and dust mixtures, the depol decreased with transport time because dust particle deposited with time. This is not our case.

The observations of PLDR in Figure 11 are all above 5 km. Observations below 5 km are filtered out to avoid results in the lower troposphere. But this information is mistakenly not introduced in the manuscript.

CALIPSO data are indeed noisy and suffer from significant error in some cases where the aerosol plumes were not correctly classified. At current stage, we decide to remove Figure 11 and the argument about depolarization ratio increasing with travel time. Still we think it might be a good chance and it is interesting to study smoke aging, but drawing convincing conclusions, much more efforts are needed in investigating and verifying CALIPSO measurements. We will continue this work in the next step.

P13, L23: Regarding the laboratory studies: Miffre et al (2016) used Arizona Test Dust (ATD). This dust is not a natural dust component. It is 'manufactored' by a company and ATD particles have rather sharp

edges. As a result, ATD shows larger depol ratios then usual desert dust particles.

Regarding the Jaervinen et al. (2016) study: These authors made their observations by one wavelength (488 nm, and partly at 552 nm) only! ..and not at 355 and 532 nm, as you mention it. But they showed their observations as a function of the size parameter (size mode diameter /wavelength) so that Mamouri and Ansmann (2016) estimated the probable depolarization ratios for 355, 532, and and 1064 nm. The results (depol ratio for fine dust at 355, 532, and 1064 nm derived from the Jaervinen study) are shown in Haarig et al. (2018). Almost all your discussion information on page 13 (L23-34) was discussed in the same way in the papers of Mamouri and Ansmann (2014, 2016). So, please add these papers to your references.

29. P14, L37: Regarding the effective radius (from the lidar inversion retrieval), please compare with Haarig et al. (2018). Is there agreement? Please discuss the comparison.

**A29**: Comparison and comments are added.

30. P14, L39: Again, please avoid mixing of tropospheric and stratospheric effects of potential particle growth. Müller et al (2007b) focusses on tropospheric smoke only when discussing aging effects.

**A30**: See **A28**.

31. P15, conclusion section: The results of the paper are summarized only in this section. But concluding remarks are not given. So maybe, create some concluding remarks, some outlook ideas...

**A31**: Modifications are made.

---

## Author Response (AR2)

List of changes:

1. One co-author added

2. Other modifications are all specific to the uncertainty estimation in the microphysical data and are highlighted in the revised manuscript.

*The paper is improved and most of the reviewers' comments have been addressed. However, one comment remains to be addressed and this is related to the uncertainties of the microphysical retrieval and the DRF impact of Canadian smoke.*

*Specifically, the fact that the inversion provides reasonable estimates for dust, does not support its applicability to smoke particles which have different refractive index and shape. Even more so, the algorithm used (i.e. Veselovskii et al. 2010) provides reasonable estimates of volume and effective radius of dust, by reproducing the depolarization measurements along with the backscatter and extinction. This is not the case presented here, since the measured depolarization values are not reproduced.*

*It is expected that the uncertainties are larger than the ones reported in the paper and that this uncertainties propagate to the DRF estimation as well. This is a serious concern that has to be addressed, otherwise the paper results are not sufficiently supported.*

*I propose that the authors will clearly state the level of uncertainties in all aspects discussed (from the inversion to DRF). This statement should be present also in Abstract and Conclusions section. The paper should stick on the EARLINET observations presented and how these can be utilized for radiative transfer applications, stating though that this is just a demonstration of future potential application of EARLINET data and techniques on DRF studies, while we are still working on the uncertainties.*

**Reply:** Actually the observed particle linear depolarization ratios can be well reproduced by spheroid model. Assume that the mixture is composed of 85% spheroids and 15% spheres, and that the complex refractive index is 1.52-i0.025. The size distribution is taken from the retrieval based on sphere model (not shown in the paper, but close to *N(0.3, 0.2)* distribution). The derived particle linear depolarization ratios are: **21% at 355 nm, 19% at 532 nm and 7.5% at 1064 nm (The real measurements are 23%, 18-19% and 5%). This simulation well reproduces the values and the spectrum of the depolarization ratio, which, to some extent, justifies the retrieved microphysical properties.**

The reason that prevented us from choosing spheroid model to invert altogether Lidar extinction, backscattering coefficient and depolarization ratio is that it tends to provide solutions with very low imaginary part as favorable solutions (because fitting the optical data with as low discrepancy as possible is the principle of inversion algorithm), while this low imaginary part is not realistic for smoke. This is the deficiency of the spheroid model.

Based on this fact, we admit that the errors of the retrieval cannot be accurately provided.

The errors we provide in Table 3 are used as a reference. The real errors could be larger or maybe smaller than those. More efforts in developing scattering model are needed.

Ansmann et al., 2018 and Haarig et al., 2018 observed smoke layers from the same event and the observation time is close to ours. They inverted their lidar data with the same algorithm. The retrieved effective radius agrees very well with our results and their retrieved volume concentration is verified by comparing with AERONET product. It is a good example that justifies the retrieval of volume concentration.

Although the errors of the complex refractive index cannot be quantitatively provided, values retrieved using sphere model (shown in the paper) are quite reasonable for smoke aerosols. In order to estimate the impact of error in the imaginary part on the DRF and heating ratio, we vary the imaginary part by +/- 50% and re-run the model. We derived up to 20% of error at the bottom of the atmosphere and up to 40% in the heating rate of the smoke plume. This estimation is based on Lille data on 24 August 2017.

The above information is summarized and added to the revised version of the manuscript.

[revised manuscript text omitted]